# Structural plasticity of actin-spectrin membrane skeleton and functional role of actin and spectrin in axon degeneration

**Guiping Wang[1,2†], David J Simon[3,4†‡], Zhuhao Wu[3], Deanna M Belsky[4], Evan Heller[1,2], Melanie K O'Rourke[4], Nicholas T Hertz[3,4], Henrik Molina[5], Guisheng Zhong[1,2], Marc Tessier-Lavigne[3,4*], Xiaowei Zhuang[1,2*]**

[1]Department of Chemistry and Chemical Biology, Howard Hughes Medical Institute, Harvard University, Cambridge, United States; [2]Department of Physics, Howard Hughes Medical Institute, Harvard University, Cambridge, United States; [3]Laboratory of Brain Development and Repair, The Rockefeller University, New York, United States; [4]Department of Biology, Stanford University, Stanford, United States; [5]Proteomics Resource Center, The Rockefeller University, New York, United States

**\*For correspondence:**
tessier3@stanford.edu (MT-L);
zhuang@chemistry.harvard.edu
(XZ)

†These authors contributed equally to this work

**Present address:** ‡Department of Biochemistry, Weill Cornell Medical College, New York, United States

**Abstract** Axon degeneration sculpts neuronal connectivity patterns during development and is an early hallmark of several adult-onset neurodegenerative disorders. Substantial progress has been made in identifying effector mechanisms driving axon fragmentation, but less is known about the upstream signaling pathways that initiate this process. Here, we investigate the behavior of the actin-spectrin-based Membrane-associated Periodic Skeleton (MPS), and effects of actin and spectrin manipulations in sensory axon degeneration. We show that trophic deprivation (TD) of mouse sensory neurons causes a rapid disassembly of the axonal MPS, which occurs prior to protein loss and independently of caspase activation. Actin destabilization initiates TD-related retrograde signaling needed for degeneration; actin stabilization prevents MPS disassembly and retrograde signaling during TD. Depletion of βII-spectrin, a key component of the MPS, suppresses retrograde signaling and protects axons against degeneration. These data demonstrate structural plasticity of the MPS and suggest its potential role in early steps of axon degeneration.
DOI: https://doi.org/10.7554/eLife.38730.001

## Introduction

Neurons and their axons are produced in excess during mammalian development, followed by a wide-spread culling of excess axons or axonal branches as the nervous system matures. This is exemplified in the development of the peripheral nervous system where sensory axons of the dorsal root ganglion (DRG) compete for limited quantities of target-derived neurotrophins that promote axon survival. Axons that receive sufficient trophic support survive and mature, whereas axons that are insufficiently supported by neurotrophins degenerate (*Luo and O'Leary, 2005*; *Schuldiner and Yaron, 2015*). Trophic deprivation (TD) triggers the axon to degenerate in three broad steps. Upon TD, the axon first generates a retrograde signal to the neuronal cell body; next, this retrograde signal initiates a pro-degenerative transcriptional response in the cell body; finally, the cell body generates an anterograde signal that propagates to the axon and gates the activation of a latent apoptotic cascade within the axon (*Ghosh et al., 2011*; *Simon et al., 2016*; *Maor-Nof et al., 2016*). A major unanswered question is to define the early signals that induce retrograde signaling to initiate axon degeneration. These upstream pathways represent attractive therapeutic targets to mitigate pathological axon degeneration (*Chao and Lee, 2004*; *Le Pichon et al., 2017*).

It has been shown recently that actin, spectrin and associated molecules form a Membrane-associated Periodic Skeleton (MPS) in axons (*Xu et al., 2013*), as well as some dendrites (*Zhong et al., 2014*; *D'Este et al., 2015*; *Han et al., 2017*), of neurons. The neuronal MPS contains molecules homologous to the components of the erythrocyte membrane skeleton (*Bennett and Baines, 2001*) but adopts a different ultrastructural organization. In the MPS, actin is arranged into ring-like structures, and these actin rings are connected by heterotetramers of α- and β-spectrin, forming a quasi-one-dimensional (1D) periodic lattice structure underneath the plasma membrane (*Xu et al., 2013*). The MPS is assembled early during axon development, forming initially in the axonal region proximal to the cell body and then propagating outward to span the entire axonal shaft (*Zhong et al., 2014*). The MPS is also found in dendrites (*Zhong et al., 2014*; *D'Este et al., 2015*) but with lower propensity and developmental rate than in axons (*Han et al., 2017*). The MPS is present in all neuronal types examined in the central and peripheral nervous systems (*D'Este et al., 2016*; *He et al., 2016*) and is evolutionarily conserved across diverse species ranging from *Caenorhabditis elegans* to *Homo sapiens* (*He et al., 2016*). Genetic perturbations of components of the MPS suggest its importance in processes such as stability and survival of axons under mechanical stress (*Hammarlund et al., 2007*), touch sensation (*Krieg et al., 2014*), the formation of axon initial segments (AIS) and nodes of Ranvier (*Voas et al., 2007*; *Zhong et al., 2014*; *D'Este et al., 2017*; *Huang et al., 2017a*), and the maintenance of axon morphology (*Leite et al., 2016*). The MPS is also found to be important for organizing some membrane proteins, such as ion channels and adhesion molecules (*Xu et al., 2013*; *Zhong et al., 2014*; *D'Este et al., 2015*; *D'Este et al., 2017*), membrane filtering at the AIS (*Albrecht et al., 2016*), and maintenance of axonal microtubules (*Qu et al., 2017*). Disruption of the MPS causes widespread neurodegeneration and a variety of neurological impairments (*Huang et al., 2017a*; *Huang et al., 2017b*) and has been implicated in neuronal toxicity in Parkinson's disease (*Ordonez et al., 2018*).

Understanding the function of the MPS requires observing and altering its behavior and functional organization in the context of specific biological processes. Here, we studied the MPS in the context of sensory axon degeneration. We found that trophic deprivation (TD) induced a rapid disassembly of the MPS, an observation also reported independently (*Unsain et al., 2018*) while our paper was in preparation. Beyond this shared observation, our data further revealed that MPS disassembly occurred early during TD-mediated axon degeneration, prior to detectable caspase activation and protein loss in axons. Importantly, MPS disassembly proceeded even when activation of the apoptotic pathway was blocked, indicating that this structural reorganization is not simply a secondary consequence of TD-dependent proteolysis but may act as a cellular signal early in TD-induced axon degeneration. Furthermore, we showed that acute pharmacological destabilization of actin, which disrupted the MPS, phenocopied TD and generated a pro-degenerative retrograde signal. In contrast, pharmacological stabilization of actin blocked MPS disassembly and prevented TD-induced retrograde signaling. Finally, and surprisingly, genetic knockdown or knockout of βII-spectrin, which inhibits the formation of the MPS (*Zhong et al., 2014*; *Han et al., 2017*), prevented retrograde signaling and protected axons from degeneration induced by both TD and actin destabilization. Together, our findings suggest a model that disassembly of the MPS and its associated proteins, which occur early after TD, may play an important role in retrograde signaling during sensory axon degeneration.

## Results

### The MPS is disassembled early upon TD

Sensory axons rely on neurotrophic factors (e.g. nerve growth factor, NGF) for their survival. During TD, multiple pathways converge to drive activation of the intrinsic apoptotic pathway and caspase-dependent cytoskeletal disassembly and axon fragmentation. Genetic deletion of pro-apoptotic proteins Bax, Caspase-9, and Caspase-3 each potently protects axons from degeneration (*Simon et al., 2012*; *Cusack et al., 2013*; *Unsain et al., 2013*). To promote degeneration, Caspase-3 cleaves a number of key substrates including Calpastatin, an endogenous inhibitor of Calpains. Together, Caspases and Calpains execute axon degeneration by cleaving many proteins including cytoskeleton components (*Yang et al., 2013*). Caspase activation is thought to be one of the early steps of axon degeneration as it precedes observable signs of degeneration such as blebbing of the plasma

membrane and fragmentation of the microtubule network (*Schoenmann et al., 2010*; *Simon et al., 2012*). The early Caspase activity can cleave proteins and induce more subtle ultrastructural changes in the axon before larger scale axon fragmentation.

We monitored the structure of the MPS during TD using a super-resolution imaging method, Stochastic Optical Reconstruction Microscopy (STORM) (*Rust et al., 2006*; *Huang et al., 2008*). We performed a time course of TD and quantified MPS structural changes by autocorrelation analysis of the 3D STORM images of βII spectrin, a key component of the MPS (*Figure 1*; *Figure 1—figure supplement 1*). Axons with a highly periodic MPS organization produce a periodic autocorrelation curve with a high amplitude, and the amplitude decreases as the structure is degraded (*Zhong et al., 2014*; *He et al., 2016*; *Han et al., 2017*). Analysis of the MPS in this study was focused on distal axons that were less bundled and exhibited a more regular MPS structure than proximal axons in DRG neurons (*Figure 1—figure supplement 2*). We initially hypothesized that MPS would be disassembled after Caspases and Calpains are active in the axon, because they cleave protein

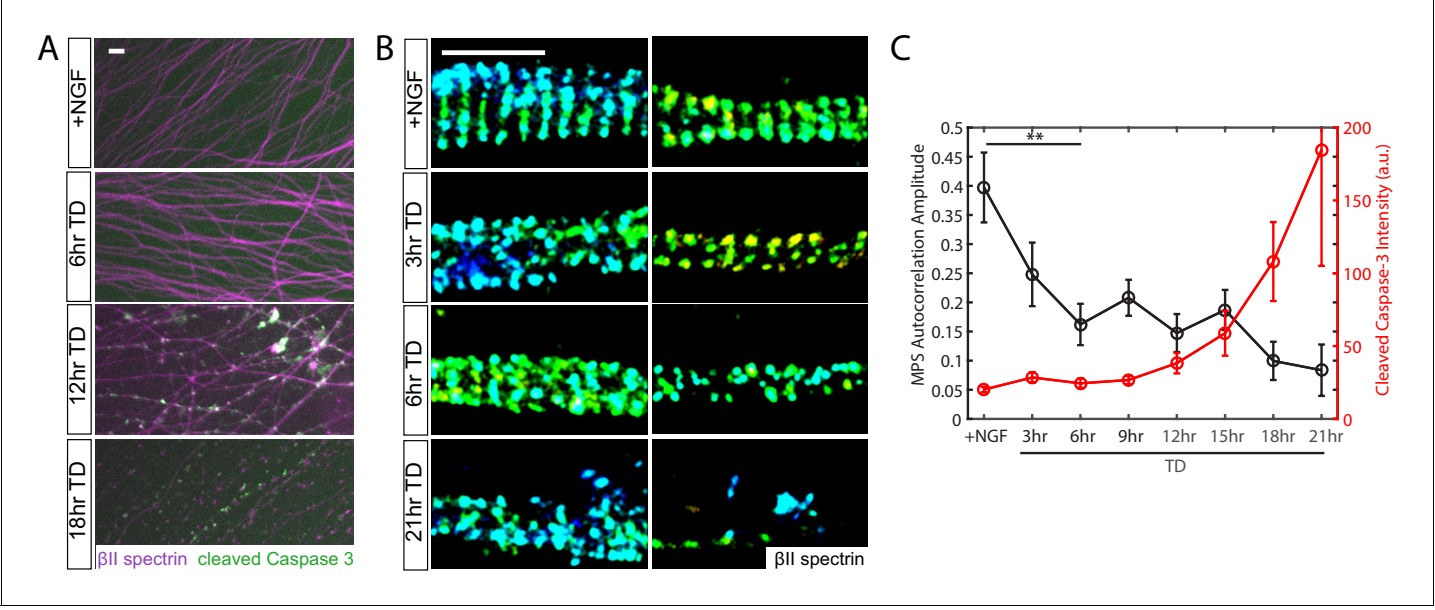

**Figure 1.** MPS disassembly upon TD precedes observable Caspase-3 activation and axon fragmentation. Dissociated and re-aggregated wildtype (WT) DRG neurons were cultured for 7 days in the presence of 50 ng/ml NGF and subsequently deprived of NGF (trophic deprivation, TD) for the indicated times. (**A**) Wide-field images of WT axons fixed after a TD time course and stained with antibodies against βII-spectrin (purple) and cleaved (activated) Caspases-3 (green). Scale bar: 20 µm. (**B**) Representative 3D STORM images of βII-spectrin in axonal regions during a TD time course. Scale bar: 1 µm. (**C**) Quantification of the degree of periodicity of βII-spectrin calculated from STORM images, and cleaved Caspase-3 intensity measured by conventional wide-field microscopy. Autocorrelation analysis is applied to each STORM image of βII-spectrin (*Figure 1—figure supplement 1*) and the autocorrelation amplitude, as defined in *Figure 1—figure supplement 1C*, provides a quantification of the degree of periodicity. Axons with highly periodic MPS produce larger autocorrelation amplitude values as compared to axons with lower degree of periodicity. The reported value is the average autocorrelation amplitude derived from many axon regions selected at random, and only a few representative examples are shown in (**B**). Statistics: data are represented as mean ± SEM. Axon number: 19–41 per condition. **p≤0.01. The p value is derived from a two-sided Kolmogorov-Smirnov test. The individual values of MPS autocorrelation amplitudes and cleaved caspase three intensity are listed in *Figure 1—source data 1*.
DOI: https://doi.org/10.7554/eLife.38730.002

The following source data and figure supplements are available for figure 1:

**Source data 1.** This spreadsheet contains the values of MPS autocorrelation amplitudes and cleaved Caspase-3 intensity used to generate *Figure 1C*.
DOI: https://doi.org/10.7554/eLife.38730.006

**Figure supplement 1.** Autocorrelation analysis of the MPS.
DOI: https://doi.org/10.7554/eLife.38730.003

**Figure supplement 2.** βII-spectrin shows substantially less periodicity in axon regions proximal to the cell body in DRG neurons.
DOI: https://doi.org/10.7554/eLife.38730.004

**Figure supplement 2—source data 1.** This spreadsheet contains the values of MPS autocorrelation amplitudes in proximal axons used to generate *Figure 1—figure supplement 2B*.
DOI: https://doi.org/10.7554/eLife.38730.005

components of the MPS such as spectrin and actin (*Yang et al., 2013*). To our surprise, we observed significant disruption of MPS within 6 hr of TD when there was neither obvious axon fragmentation nor detectable increase in immunoreactivity for cleaved (activated) Caspase-3 (*Figure 1*).

## The early disassembly of MPS is independent of apoptotic pathway activation and protein loss

We next investigated whether this early MPS disassembly could result from low-level activation of the apoptotic pathway that could not be observed within the detection limits of our assays. We therefore repeated the TD time course in sensory axons that overexpressed the anti-apoptotic protein Bcl-xL. Bcl-xL inhibits Bax, which is in turn required to activate effector Caspases, and it has been shown that the overexpression of Bcl-xL potently blocks axon degeneration following TD (*Figure 2A*) (*Vohra et al., 2010*; *Simon et al., 2016*). Notably, we found that overexpression of Bcl-xL did not inhibit the early disruption of the MPS upon TD (*Figure 2B,C*). In parallel, we also measured changes in the MPS in axons from *Bax*-knockout mice (*Knudson et al., 1995*), which are similarly protected from degeneration following TD (*Figure 2A*) (*Nikolaev et al., 2009*; *Simon et al., 2012*). Consistent with our Bcl-xL overexpression data, MPS disassembly proceeded normally in *Bax*-knockout axons (*Figure 2D,E*). Notably, while TD causes a substantial decrease in MPS periodicity, the average fluorescence intensity of βII spectrin in these axon regions was unchanged during the first 12 hr of TD (*Figure 2—figure supplement 1*), suggesting that the disassembly of the MPS was not a result of the reduced abundance of its protein components during TD. In addition, we repeated a TD time course in parallel cultures and analyzed axon-specific protein lysates by immunoblotting. The overall levels of αII spectrin, βII spectrin and actin in axons remained constant during the first 9 hr of TD when MPS disassembly already started (*Figure 2F*), and only at later time points did we observe cleavage of αII spectrin, reflective of a Caspase-dependent cleavage of this protein during TD (*Yang et al., 2013*). Indeed, this late cleavage of αII spectrin was blocked by expression of Bcl-xL. Together these results indicate that early MPS disassembly upon TD is not a result of activation of the apoptotic pathway or the related degradation of the MPS protein components.

As part of an ongoing, large-scale, unbiased phosphoproteomic screen in axons undergoing TD, we discovered that some protein components of the MPS, including spectrin and adducin, were widely phosphorylated during TD (*Figure 2—figure supplement 2*, *Figure 2—figure supplement 2—source data 1*). While the full impact of these phosphorylation sites on the MPS components remains to be investigated, analogous phosphorylations on the short C-terminal isoform of βII spectrin have been previously shown to reduce interactions with αII spectrin (*Bignone et al., 2007*) and thereby potentially inhibit the formation of the spectrin tetramer. Additionally, analogous phosphorylation on adducin, a protein known to enhance actin-spectrin interaction (*Bennett and Baines, 2001*), has been found to inhibit formation of spectrin-actin complexes and to affect spectrin localization (*Matsuoka et al., 1996*; *Matsuoka et al., 1998*). Therefore, it is possible that the pattern of phosphorylation that we detected on MPS components upon TD may facilitate dissociation of the spectrin-actin-adducin complex and contribute to MPS disassembly.

## Pharmacological actin destabilization induces MPS disassembly and pro-degenerative retrograde signals

We have previously shown that latrunculin A (LatA), which binds actin monomers and prevents their polymerization, disrupts the MPS in hippocampal neurons (*Xu et al., 2013*; *Zhong et al., 2014*; *Han et al., 2017*). Consistent with our previous results, 3 hr incubation with LatA at a concentration of 20 μM was sufficient to completely disassemble the axonal MPS in DRG sensory neurons and this effect lasted for up to 12 hr after LatA removal (*Figure 3A,B*).

We next asked whether the LatA treatment, which causes MPS disassembly, might induce axon degeneration in a manner similar to TD. One of the earliest steps by which TD causes axon degeneration is through the generation of retrograde signals from the axon to the cell body, resulting in phosphorylation and activation of key downstream transcription factors including c-Jun (*Ghosh et al., 2011*; *Simon et al., 2016*). Retrograde signaling functions upstream of caspase activation and depends critically on multiple kinases including the MAPKKK Dual Leucine Zipper Kinase (DLK) (*Ghosh et al., 2011*). During TD, DLK activity promotes phosphorylation and activation of c-Jun (*Xu et al., 2001*; *Ghosh et al., 2011*; *Huntwork-Rodriguez et al., 2013*; *Larhammar et al.,*

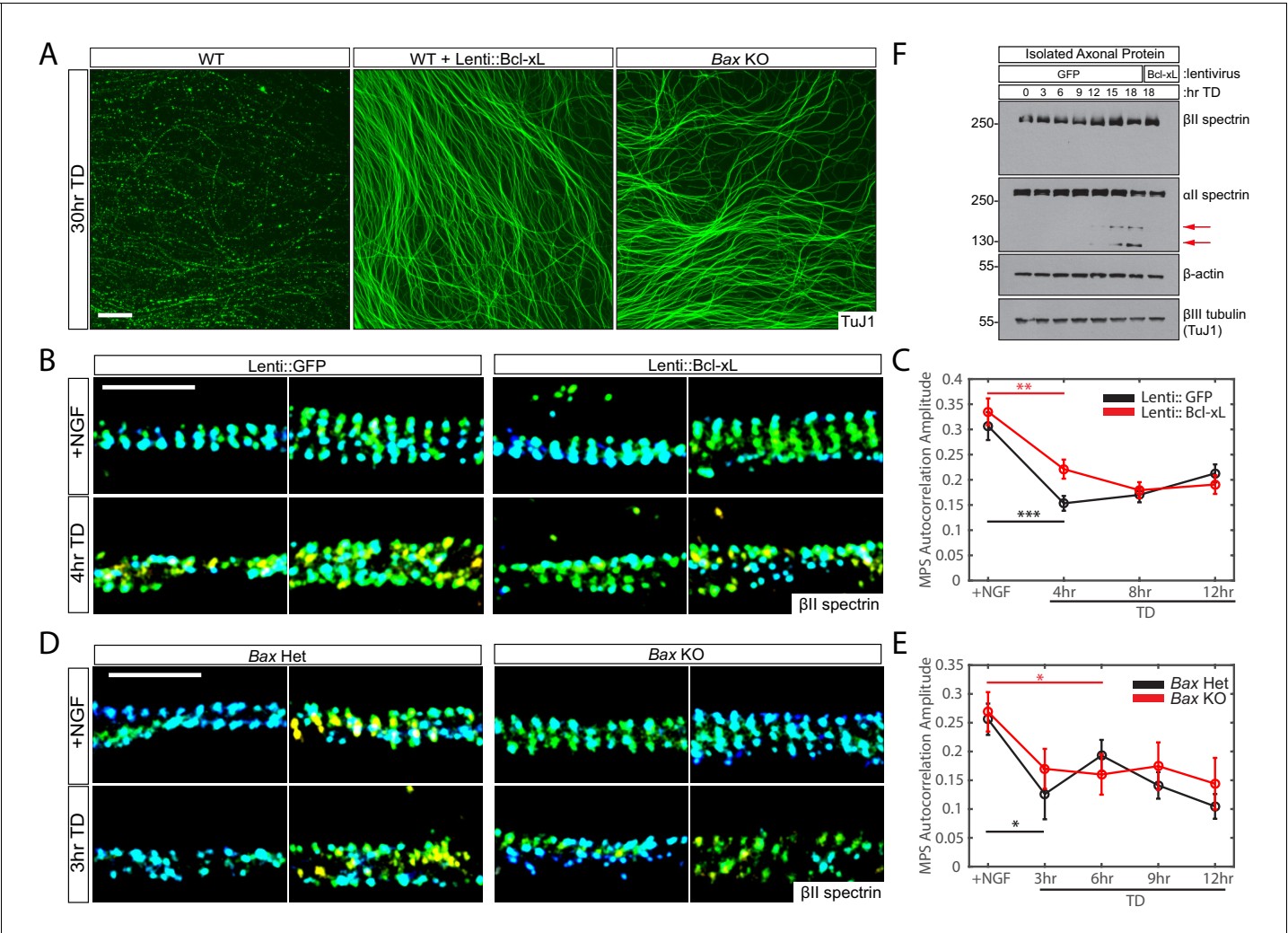

**Figure 2.** The disassembly of MPS upon TD is independent of apoptotic pathway activation. (**A**) Representative images of 7-day in vitro (DIV) DRG cultures from indicated genotypes and treatments (described as below) following 30 hr TD. Axons are visualized with an antibody to βIII Tubulin (TuJ1). Scale bar: 100 µm. (**B and C**) Dissociated and re-aggregated WT DRG cultures were transduced with lentivirus expressing GFP (B-left panels, C-black curve) or Bcl-xL (B-right panels, C-red curve) one day after plating and deprived of NGF (TD) for varying durations beginning at 8 DIV. Representative 3D STORM images of βII-spectrin in axonal regions in the presence of NGF (top) or at 4 hr of TD (bottom) are shown in (**B**). Scale bar: 1 µm. βII-spectrin autocorrelation amplitude values during the TD time course are quantified in (**C**). Statistics: Data are represented as mean ± SEM. Axon number: 66–104 per condition. **p≤0.01; ***p≤0.001. p-Values are derived from a two-sided Kolmogorov-Smirnov test. The individual values of MPS autocorrelation amplitudes are listed in *Figure 2—source data 1*. (**D and E**) Dissociated and re-aggregated DRG cultures from *Bax* heterozygous (Het, D-left panels, E-black curve) or homozygous knockout (KO, D-right panels, E-red curve) embryos were deprived of NGF (TD) for varying durations starting at 7 DIV. Representative 3D STORM images of βII-spectrin in axonal regions in the presence of NGF (top) or at 3 hr of TD (bottom) are shown in (**D**). Scale bar: 1 µm. Quantification of autocorrelation amplitude values for βII-spectrin during the TD time course is shown in (**E**). Statistics: Data are represented as mean ± SEM. Axon number: 27–53 per condition. *p≤0.05. p-Values are derived from a two-sided Kolmogorov-Smirnov test. The individual values of MPS autocorrelation amplitudes are listed in *Figure 2—source data 1*. (**F**) Protein lysates of isolated axons from 8 DIV DRG cultures expressing the indicated lentiviral constructs (GFP or Bcl-xL) were generated at the indicated time points of TD. Immunoblotting was performed using the indicated antibodies. Red arrows indicate the appearance of degeneration-associated cleavage products of αII spectrin.

DOI: https://doi.org/10.7554/eLife.38730.007

The following source data and figure supplements are available for figure 2:

**Source data 1.** This spreadsheet contains the values of MPS autocorrelation amplitudes used to generate *Figure 2C,E*.
DOI: https://doi.org/10.7554/eLife.38730.012

**Figure supplement 1.** TD does not induce global reduction in axonal βII-spectrin abundance during the first 12 hr of TD.
DOI: https://doi.org/10.7554/eLife.38730.008

**Figure supplement 1—source data 1.** This spreadsheet contains the values of fluorescence intensity used to generate *Figure 2—figure supplement 1*.

*Figure 2 continued on next page*

*Figure 2 continued*

DOI: https://doi.org/10.7554/eLife.38730.009

**Figure supplement 2.** MPS components αII-spectrin, βII-spectrin, and adducin are widely phosphorylated during TD.
DOI: https://doi.org/10.7554/eLife.38730.010

**Figure supplement 2—source data 1.** This spreadsheet contains the information of the phosphorylation sites shown in *Figure 2—figure supplement 2*.
DOI: https://doi.org/10.7554/eLife.38730.011

*2017*). Interestingly, in the context of axon injury, retrograde signaling by activated DLK can also promote axon regeneration (*Hammarlund et al., 2009*; *Yan et al., 2009*; *Shin et al., 2012*; *Watkins et al., 2013*).

Activation of DLK during TD is associated with an increase in its apparent molecular weight on a western blot, reflective of TD-associated phosphorylations (*Huntwork-Rodriguez et al., 2013*). Similar to the case of TD, we observed an increase in the molecular weight of DLK in axons upon LatA treatment (*Figure 3—figure supplement 1*). In addition, we found that acute application of LatA strongly increased c-Jun phosphorylation and this effect was blocked by co-addition of a small molecule DLK inhibitor, GNE-3511, also similar to the case of TD (*Figure 3C,D*). Similar activation of c-Jun in sensory axons has been observed following acute disruption of actin and microtubules with cytochalasin D and nocodazole, respectively (*Valakh et al., 2015*).

It has been shown that retrograde signaling induces a transcriptional response in the cell body and the generation of an anterograde signal from the cell body to the axon that activates a latent pool of Caspase-3 in the axon. Activation of axonal Caspase-3 in turn triggers axon fragmentation (*Ghosh et al., 2011*; *Simon et al., 2016*; *Maor-Nof et al., 2016*). Indeed, physically isolating the axon from its cell body blocks the anterograde signaling and hence the TD-induced axonal Caspase-3 activation (*Simon et al., 2016*). We therefore examined whether LatA treatment activates axonal Caspase-3, and whether this process, as in the case of TD, requires anterograde signaling from the cell body by severing the axons from their cell bodies immediately prior to LatA treatment. To enable this experiment, we expressed a cytoplasmic variant of the biosynthetic enzyme NMNAT1 (cytoNMNAT1) in wild-type axons, a widely used tool that blocks injury-induced Wallerian degeneration of the axon without interfering with Caspase-3 activation during TD-induced axon degeneration (*Sasaki et al., 2009*; *Simon et al., 2016*). Notably, we found that application of LatA, like TD, induced an increase in cleaved (activated) Caspase-3 in intact axons but not in axons severed from the cell bodies (*Figure 3E*). In addition, we also observed an increase in Caspase-3-like enzymatic activity ('DEVDase') in intact axons, but not in isolated axons, in a way similar to TD, although the magnitude of TD-induced axonal DEVDase activity was consistently greater than that elicited by LatA (*Figure 3F*). These findings indicate that LatA treatment, like TD, does not directly activate the apoptotic pathway in axons, but instead activates a somatic pro-degenerative pathway that in turn activates axonal Caspase-3 via an anterograde signaling pathway from the cell body.

## Pharmacological actin stabilization preserves MPS during TD and inhibits pro-degenerative retrograde signals

Next, we asked whether stabilizing F-actin had the opposite effect on TD-induced MPS disassembly and retrograde signaling. We used Jasplakinolide (Jasp), a toxin that promotes actin polymerization and thereby stabilizes F-actin. We note that the addition of Jasp occasionally caused depletion of the MPS in a fraction of axonal segments, but the majority of axonal regions exhibited intact MPS (*Figure 4—figure supplement 1*). We repeated a time course of TD in the presence of Jasp and found that Jasp effectively prevented disassembly of the MPS upon TD (*Figure 4A,B*). This observation is consistent with the independent finding that actin stabilization can inhibit MPS disassembly and axon fragmentation (*Unsain et al., 2018*), but TD-induced retrograding signaling has not been studied in *Unsain et al. (2018)*. To examine the effect of Jasp treatment on TD-induced retrograde signaling in axon degeneration, we examined phosphorylation of c-Jun and found that application of Jasp prevented the TD-induced rise in phosphorylated c-Jun (p-c-Jun) (*Figure 4C,D*), which is required in part for TD-induced degeneration (*Simon et al., 2016*). Likewise, Jasp treatment prevented the TD-induced increase in the molecular weight of axonal DLK (*Figure 4—figure supplement 2*), a proxy for DLK activation (*Huntwork-Rodriguez et al., 2013*). Interestingly, application of

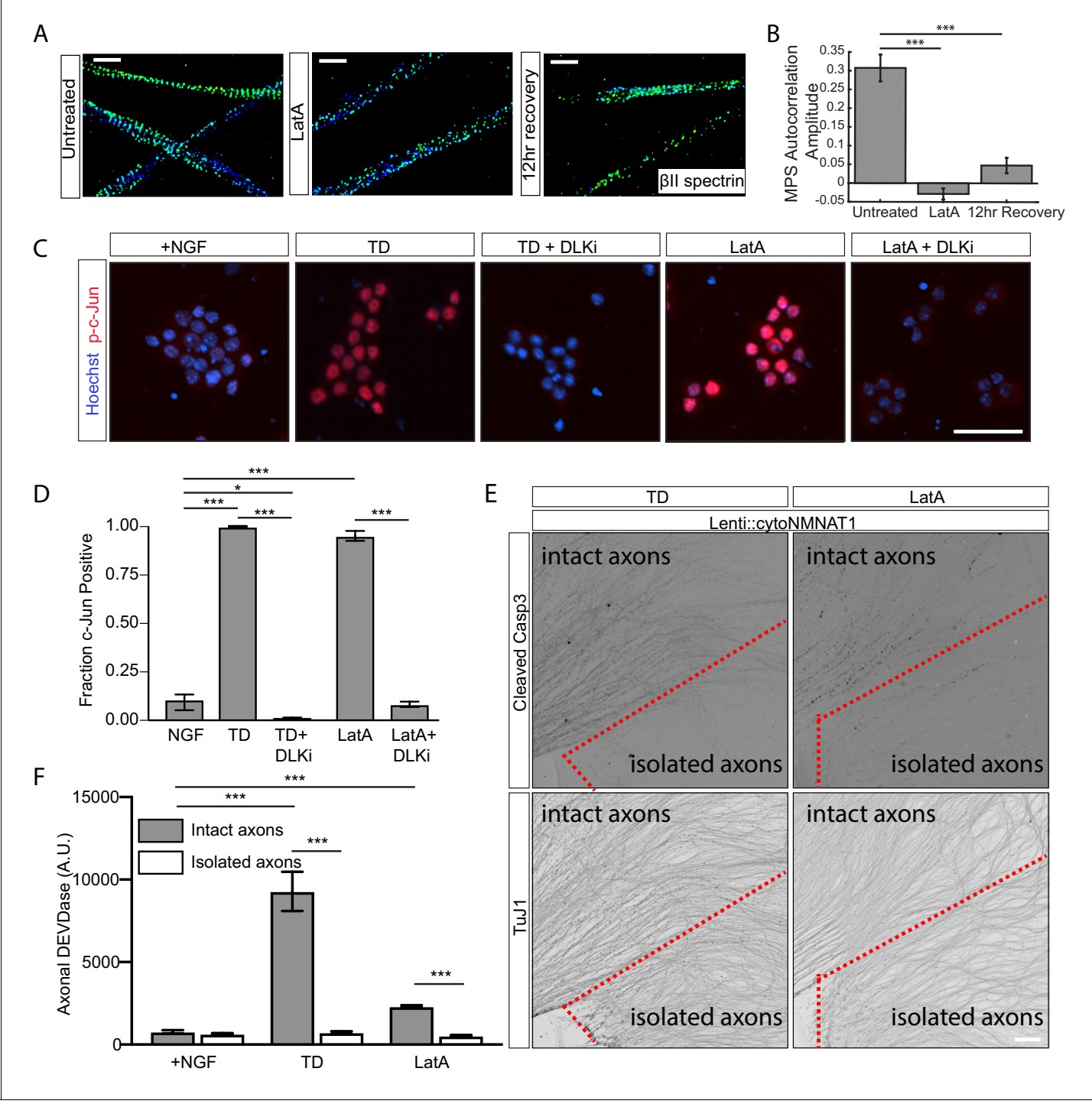

**Figure 3.** Depolymerization of actin filaments disrupts MPS and induces pro-degenerative signals. 20 μM of latrunculin A (LatA) or vehicle was applied to relevant DRG cultures (7 DIV) for 3 hr in the presence of NGF. (**A, B**) Representative 3D STORM images of βII-spectrin in axonal regions in untreated (left), LatA-treated (middle), 12-hr recovery (cultured for an extra 12 hr after LatA removal, right) cultures are shown in (**A**) and quantified in (**B**). Scale bar: 1 μm. Statistics: Data are represented as mean ± SEM. Axon number: 27–36 per condition. ***p≤0.001. p-Values are derived from a two-sided Kolmogorov-Smirnov test. The individual values of MPS autocorrelation amplitudes are listed in *Figure 3—source data 1*. (**C**) 7 DIV DRG cultures were subjected to 6 hr TD or 6 hr treatment with 20 μM LatA, either alone or in the presence of 1 μM DLK inhibitor (DLKi) GNE-3511. Cultures were stained with the DNA dye Hoechst 33342 (blue) and an antibody to phosphorylated c-Jun (Serine 73) (p-c-Jun) (red). Cell bodies are visualized. Scale bar: 50 μm. (**D**) Quantification of the fraction of phosphorylated c-Jun positive cell bodies in (**C**). Data are represented as mean ± SEM. N = 3 independent experiments. *p≤0.05, ***p≤0.001, two-way ANOVA with Bonferroni post-test. The individual values of the fraction of phosphorylated c-Jun positive

*Figure 3 continued on next page*

*Figure 3 continued*

cell bodies are listed in *Figure 3—source data 2*. (E) Cultures of WT DRGs were established that expressed lentiviral cytoplasmic NMNAT1. A subset of axons were physically severed from their cell bodies (isolated axons, locate at the lower half of each field of view and are encircled with a dotted red line) while the rest of axons (intact axons, upper left of each field of view) remain connected to cell bodies, followed by either 12 hr TD (left) or 12 hr application of 20 µM LatA. Cultures were stained for cleaved Caspase-3 (top) to indicated Caspase-3 activity and βIII tubulin (TuJ1, bottom) to indicate axons. Scale bar: 100 µm. (F) In parallel cultures with (E), at 8 DIV, axons were either severed away from their cell bodies ('isolated axons') or left intact ('intact axons') and subjected to 16 hr of TD or 16 hr treatment with 20 µM LatA in the presence of NGF. At the end of the assay (16 hr time point), cell bodies were removed from the 'intact axons,' leaving all cultures with axons but no cell bodies. At the point, axonal DEVDase activity was measured. Data are represented as mean ± SEM. N = 3 independent experiments. ***p≤0.001, two-way ANOVA with Bonferroni post-test. The individual values of axonal DEVDase activities are listed in *Figure 3—source data 3*.
DOI: https://doi.org/10.7554/eLife.38730.013

The following source data and figure supplement are available for figure 3:

**Source data 1.** This spreadsheet contains the values of MPS autocorrelation amplitudes used to generate *Figure 3B*.
DOI: https://doi.org/10.7554/eLife.38730.015
**Source data 2.** This spreadsheet contains the values of fraction of p-c-Jun(+) cells used to generate *Figure 3D*.
DOI: https://doi.org/10.7554/eLife.38730.016
**Source data 3.** This spreadsheet contains the values of axonal DEVDase used to generate *Figure 3F*.
DOI: https://doi.org/10.7554/eLife.38730.017
**Figure supplement 1.** Latrunculin A treatment induces activation of DLK mimicking TD.
DOI: https://doi.org/10.7554/eLife.38730.014

Jasp alone caused a small rise in the amount of detectable phosphorylated c-Jun-positive cells (*Figure 4C,D*), perhaps reflecting a low-level stress response initiated by inhibition of normal actin turnover.

Together, our observations of the early caspase-independent MPS disassembly upon TD and of the correlation between the bi-directional modulation of MPS stability and retrograde signaling in axon degeneration suggest a potential signaling role for this axonal membrane skeleton in the initiation of axon degeneration.

## Depletion of the MPS component βII spectrin protects axons from degeneration

Next, we investigated the effect of depletion of βII spectrin, a key component of the MPS, on TD-induced axon degeneration. To achieve this, we transfected DRG neurons with a validated adenovirus (AV)-based shRNA to βII spectrin (*Sptbn1*) (*Galiano et al., 2012*; *Zhong et al., 2014*; *Han et al., 2017*) and observed nearly complete removal of βII spectrin at the time of our assay (*Figure 5—figure supplement 1A*). Axons derived from these cultures displayed a down-regulation of some additional MPS components (αII spectrin and ankyrin-B), perhaps reflective of a stabilizing effect of the MPS against the degradation of some of its components, though βII spectrin depletion caused no change in the abundance of actin and βIII tubulin (*Figure 5—figure supplement 2*). Remarkably, we found that knockdown of βII spectrin strongly protected axons from TD-induced degeneration (*Figure 5A,B*). TD is known to induce a rise in Caspase-3-like enzymatic activity ('DEVD-ase') in axons (*Simon et al., 2016*), and we found that βII spectrin knockdown also blocked this rise in 'DEVD-ase' activity upon TD (*Figure 5—figure supplement 1B*). To further validate this finding, we established cultures from βII spectrin conditional knockout mice and infected these cultured neurons with Cre recombinase via lentiviral constructs to achieve a comparable removal of βII spectrin. Consistent with our shRNA knockdown results, genetic deletion of βII spectrin also protected axons from degeneration following TD (*Figure 5C,D*). In contrast to the protection observed against TD, loss of βII spectrin offered no appreciable protection against axon degeneration caused by direct activation of the apoptotic pathway (by the Bcl-2/Bcl-xL/Bcl-w antagonist ABT-737), or direct damage to the mitochondria (by rotenone) or to the microtubule cytoskeleton (by vincristine) (*Figure 5—figure supplement 1C*). We next examined activation of c-Jun and Foxo3a in wild type and βII-spectrin-deficient axons undergoing TD. We found that βII-spectrin knockdown inhibited the TD-induced rise in c-Jun mRNA (*Figure 5—figure supplement 1D*), and prevented the increase in phosphorylated c-Jun (*Figure 5E,F*) and nuclear-localized Foxo3a (*Figure 5G,H*) upon TD, which are early events during axon degeneration signaling (*Ghosh et al., 2011*; *Maor-Nof et al., 2016*; *Simon et al.,*

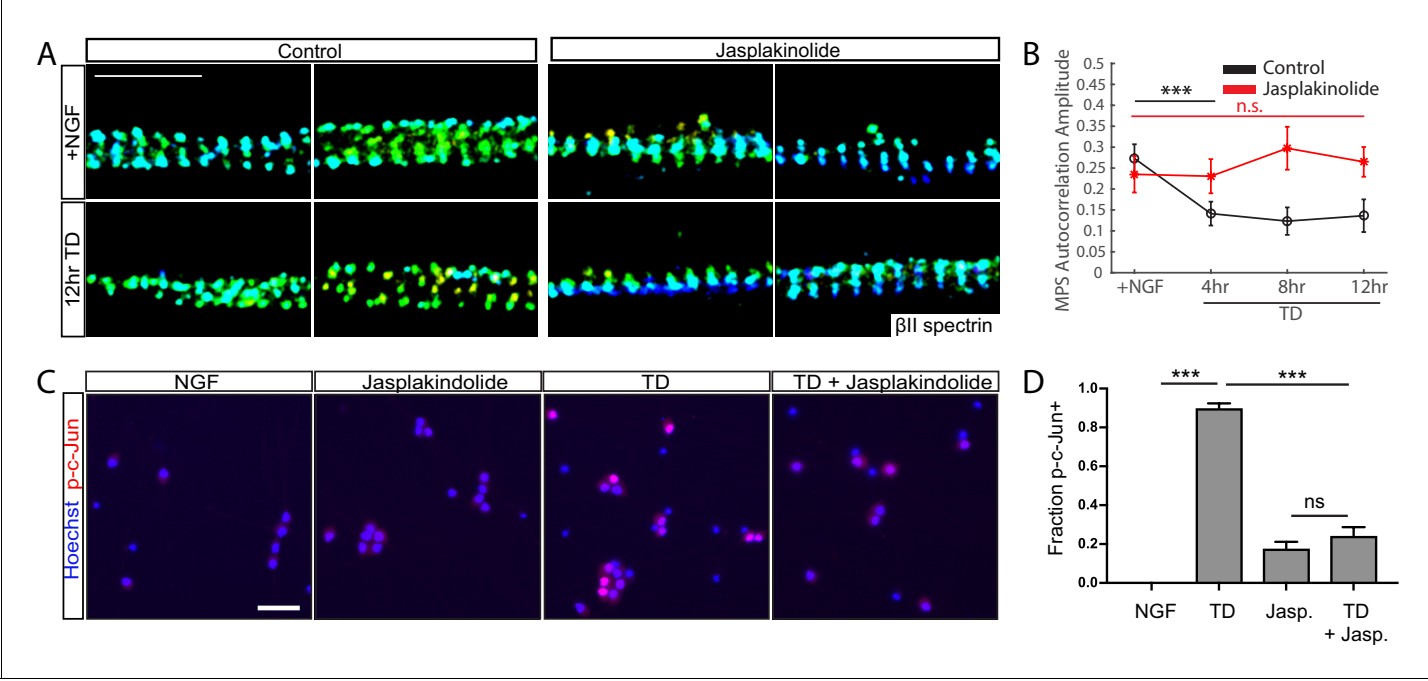

**Figure 4.** Stabilizing actin filaments preserves MPS and blocks pro-degenerative retrograde signals. WT DRG cultures (7 DIV) were pretreated with 10 µM Jasplakinolide (Jasp) or vehicle for 30 min, followed by a TD time course in the continued presence of Jasp. (A) Representative 3D STORM images of βII-spectrin in axonal regions in cultures without (left) or with (right) Jasp treatment in the presence of NGF (top) or at 12 hr of TD (bottom). Scale bar: 1 µm. (B) Autocorrelation amplitude values for βII-spectrin in cultures without (black) or with (red) application of Jasp. Statistics: Data are represented as mean ± SEM. Axon number: 39–61 per condition. ***p≤0.001; n.s. p>0.05. p-Values are derived from a two-sided Kolmogorov-Smirnov test. The individual values of MPS autocorrelation amplitudes are listed in *Figure 4—source data 1*. (C) Immunoreactivity for phosphorylated c-Jun (p-c-Jun) in 7 DIV DRG cultures in the presence of NGF or following an 8 hr TD, without or with 1 µM Jasp pre-incubation for 30 min prior to the assay, as indicated. Scale bar: 50 µm. (D) Quantification of the fraction of phosphorylated c-Jun positive cell bodies. Statistics: Data are represented as mean ± SEM. N = 3 independent experiments,~400 cell neurons per assay. ***p≤0.001. The individual values of the fraction of phosphorylated c-Jun positive cell bodies are listed in *Figure 4—source data 2*.

DOI: https://doi.org/10.7554/eLife.38730.018

The following source data and figure supplements are available for figure 4:

**Source data 1.** This spreadsheet contains the values of MPS autocorrelation amplitudes used to generate *Figure 4B*.
DOI: https://doi.org/10.7554/eLife.38730.022

**Source data 2.** This spreadsheet contains the values of fraction of p-c-Jun(+) cells used to generate *Figure 4D*.
DOI: https://doi.org/10.7554/eLife.38730.023

**Figure supplement 1.** Jasplakinolide treatment stabilizes MPS upon TD but removes MPS from a fraction of DRG axonal segments.
DOI: https://doi.org/10.7554/eLife.38730.019

**Figure supplement 1—source data 1.** This spreadsheet contains the values of occupancy ratio used to generate *Figure 4—figure supplement 1*.
DOI: https://doi.org/10.7554/eLife.38730.020

**Figure supplement 2.** Jasplakinolide treatment occludes the TD-induced activation of axonal DLK.
DOI: https://doi.org/10.7554/eLife.38730.021

*2016*). Interestingly, the TD-induced activation of DLK, an earlier step in pro-degenerative retrograde signaling was unaffected in βII-spectrin knockdown cultures (*Figure 5I*). These results suggest that the protection against axon degeneration offered by βII-spectrin removal occurs early in the degeneration pathway, upstream of axonal apoptotic pathway activation and the activation of transcription factors c-Jun and Foxo3a, but downstream of DLK phosphorylation/activation.

To examine a role for βII spectrin in pro-degenerative retrograde signaling more broadly, we applied LatA to βII-spectrin knockdown cultures and observed that loss of βII spectrin prevents the LatA-induced appearance of axonal DEVDase activity (*Figure 5—figure supplement 3A*), although interestingly not the LatA-induced rise in phospho-c-Jun (*Figure 5—figure supplement 3B,C*). This is likely because LatA treatment, which disrupts all forms of F-actin, initiates an additional, less

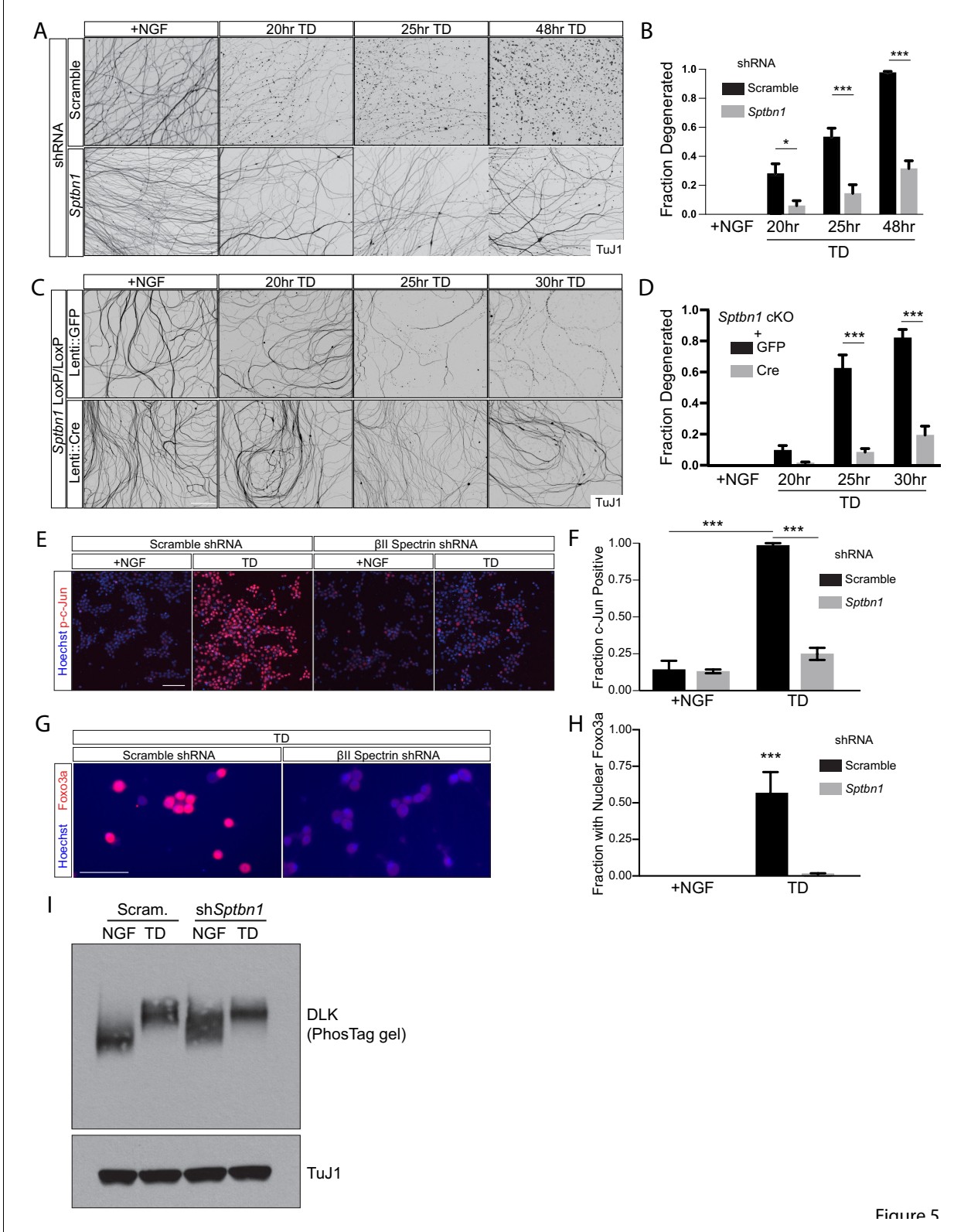

**Figure 5.** Depletion of the MPS component βII spectrin inhibits retrograde signaling and protects axons from degeneration. (**A**, **B**) Dissociated and re-aggregated WT DRG cultures were infected with AV-shRNA against βII-spectrin (*Sptbn1*) or scrambled control at 1 DIV and deprived of NGF (TD) for the indicated time points starting at 8 DIV. Axons are visualized with βIII tubulin (TuJ1) immunostaining in (**A**) and degenerated axons are quantified over time in (**B**). Scale bar: 100 μm. Statistics: n = 4 independent experiments. *p≤0.05, ***p≤0.001, two-way ANOVA with Bonferroni post-test. The

*Figure 5 continued on next page*

*Figure 5 continued*

individual values of ratio of axons degenerated are listed in *Figure 5—source data 1*. (C, D) Dissociated and re-aggregated DRG cultures from the indicated genotype were transduced with lentivirus expressing GFP as a control or lentivirus expressing Cre recombinase to delete the βII-spectrin (*Sptbn1*) gene from 1 DIV. The cultures were deprived of NGF (TD) for the indicated time points starting at 8 DIV. Axon degeneration was visualized in (C) and quantified over time in (D). Scale bar: 100 µm. Statistics: Data are represented as mean ± SEM. n = 3 independent experiments. ***p≤0.001, two-way ANOVA with Bonferroni post-test. The individual values of ratio of axons degenerated are listed in *Figure 5—source data 2*. (E) DRG cultures expressing AV-based scrambled shRNA or shRNA against βII-spectrin were subjected to 6 hr TD and stained with an antibody to phosphorylated c-Jun (p-c-Jun). Scale bar: 50 µm. (F) Quantification of the fraction of cells immunoreactive for p-c-Jun from (E). Data are represented as mean ± SEM. n = 3 independent experiments. ***p≤0.001, two-way ANOVA with Bonferroni post-test. The individual values of the fraction of phosphorylated c-Jun-positive cell bodies are listed in *Figure 5—source data 3*. (G) As in (E) but for cultures stained with antibody to Foxo3a. Nuclear import of Foxo3a is visualized by co-localization with the DNA dye Hoechst 33342. (H) Quantification of the fraction of cells from (G) where Foxo3a immunoreactivity appears to co-localize with Hoechst 33342 staining. Data are represented as mean ± SEM. n = 3 independent experiments. ***p≤0.001, two-way ANOVA with Bonferroni post-test. The individual values of the fraction of cells with nuclear Foxo3a are listed in *Figure 5—source data 4*. (I) Cultures were infected at 1 DIV with AVs expressing scrambled shRNA or shRNA against βII spectrin (*Sptbn1*). At 8 DIV, cultures were subjected to TD (12 hr), followed by removal of cell bodies and axon-specific lysis. Samples were subjected to immunoblotting as indicated. The TD-associated shift in DLK molecular weight was visualized using a phos-Tag gel (see Materials and methods). Representative western blot from three independent biological replicates.

DOI: https://doi.org/10.7554/eLife.38730.024

The following source data and figure supplements are available for figure 5:

**Source data 1.** This spreadsheet contains the values of fraction of axons degenerated used to generate *Figure 5B*.
DOI: https://doi.org/10.7554/eLife.38730.032
**Source data 2.** This spreadsheet contains the values of fraction of axons degenerated used to generate *Figure 5D*.
DOI: https://doi.org/10.7554/eLife.38730.033
**Source data 3.** This spreadsheet contains the values of fraction of p-c-Jun(+) cells used to generate *Figure 5F*.
DOI: https://doi.org/10.7554/eLife.38730.034
**Source data 4.** This spreadsheet contains the values of fraction of cells with nuclear Foxo3a used to generate *Figure 5H*.
DOI: https://doi.org/10.7554/eLife.38730.035
**Figure supplement 1.** Loss of βII spectrin occludes increase in Caspase-3/7-like enzymatic activities during TD but offers no appreciable protection against apoptosis induced by various drugs.
DOI: https://doi.org/10.7554/eLife.38730.025
**Figure supplement 1—source data 1.** This spreadsheet contains the values of axonal DEVDase used to generate *Figure 5—figure supplement 1B*.
DOI: https://doi.org/10.7554/eLife.38730.026
**Figure supplement 1—source data 2.** This spreadsheet contains the values of fold induction c-Jun mRNA used to generate *Figure 5—figure supplement 1D*.
DOI: https://doi.org/10.7554/eLife.38730.027
**Figure supplement 2.** βII spectrin depletion destabilizes some other MPS components.
DOI: https://doi.org/10.7554/eLife.38730.028
**Figure supplement 3.** Loss of βII spectrin prevents the LatA-induced appearance of axonal DEVDase activity but not LatA-induced rise in phospho-c-Jun.
DOI: https://doi.org/10.7554/eLife.38730.029
**Figure supplement 3—source data 1.** This spreadsheet contains the values of axonal DEVDase used to generate *Figure 5—figure supplement 3A*.
DOI: https://doi.org/10.7554/eLife.38730.030
**Figure supplement 3—source data 2.** This spreadsheet contains the values of fraction of p-c-Jun(+) cells used to generate *Figure 5—figure supplement 3C*.
DOI: https://doi.org/10.7554/eLife.38730.031

specific and βII-spectrin-independent signal to activate c-Jun, although this signal does not appear to lead to axon degeneration.

## Discussion

### The axonal MPS is a plastic structure that disassembles early during axon degeneration and independently of the apoptotic pathway

The membrane skeleton structure MPS, comprised of actin, spectrin and associated molecules, is assembled early during axon development and remains a stable structure in mature, unstimulated

neurons (*Zhong et al., 2014*). Here, we report data that suggest a novel plasticity in the MPS upon trophic deprivation (TD).

We show that removal of neurotrophic support causes a rapid disassembly of the MPS structure. A similar observation has been reported independently (*Unsain et al., 2018*) while our paper was in preparation, but (*Unsain et al., 2018*) did not investigate the relationship of MPS disassembly to either the Caspase-dependent apoptotic pathway or the retrograde signaling induced by TD. Several MPS components are themselves Caspase substrates, and we and others have observed low-level Caspase activation that precedes apparent signs of axon fragmentation (*Schoenmann et al., 2010*; *Simon et al., 2012*), thus the MPS disassembly could potentially be a simple consequence of the activation of the axonal apoptotic pathway during axon degeneration. However, unexpectedly, we find that MPS disassembly occurs without a change in the abundance of its component proteins. Additionally, the disassembly still proceeds normally in axons deficient in apoptotic pathway activation, suggesting that the disassembly happens upstream of Caspase-dependent proteolysis. Together these findings indicate that MPS disassembly is an early event in the TD-induced degeneration pathway, independent of Caspase-dependent proteolysis.

We also observe extensive TD-induced phosphorylation on MPS components, spectrin and adducin. Increased phosphorylation of β spectrin has been reported to decrease membrane mechanical stability in erythrocytes (*Manno et al., 1995*). In addition, analogous phosphorylation on a βII-spectrin isoform reduces its interaction with αII spectrin (*Bignone et al., 2007*) and analogous phosphorylation on adducin inhibits formation of the spectrin-actin complex (*Matsuoka et al., 1996*; *Matsuoka et al., 1998*). While the biological significance of TD-induced phosphorylation on spectrin and adducin in axons remains to be determined, it is possible that these phosphorylation events could potentially facilitate MPS disassembly by affecting the stability of the actin-spectrin complex. Furthermore, MPS disassembly upon TD is prevented by pharmacologically stabilizing F-actin, suggesting that TD-induced activation may activate one or more actin modifying enzymes, which may in turn contribute to driving the observed MPS structural changes.

## Depletion of βII-spectrin inhibits retrograde signaling and protects axons from degeneration

Loss or mutation of α/β spectrin family members from birth has profound consequences on nervous system development and integrity from *C. elegans* through mammals (*Tang et al., 2003*; *Pielage et al., 2005*; *Hammarlund et al., 2007*; *Voas et al., 2007*; *Zhang et al., 2013*; *Krieg et al., 2014*; *Krieg et al., 2017*; *Huang et al., 2017a*; *Huang et al., 2017b*). Because of the known importance of spectrin in development, to investigate its role in axon degeneration by genetic manipulation, we removed βII spectrin from established neuronal cultures using virally delivered shRNA or Cre recombinase. Our data uncover a novel post-developmental role for βII spectrin that is required for axon degeneration at an early step, likely through the regulation of retrograde signaling. We observed that the depletion of βII spectrin protects axons from degeneration following TD, and that this effect is downstream of the activation of DLK but upstream of c-Jun activation. In addition, loss of βII spectrin also inhibits the pro-degenerative effects of acute F-actin breakdown by the actin depolymerization drug LatA. It has been shown previously that beta-II spectrin depletion disrupts the MPS in cultured hippocampal neurons based on actin and adducin imaging (*Zhong et al., 2014*; *Han et al., 2017*). Because the longitudinal actin filaments mask the periodic structure more strongly and that adducin signal is relatively weak in distal axonal regions in DRG neurons, it is challenging to use these markers to assess MPS integrity in DRG neurons. However, we observed that beta-II spectrin knockdown led to substantial decrease in the abundance of some other MPS components, such as alpha-II spectrin and ankyrin-B, suggesting that the MPS is likely also disrupted in beta-II spectrin depleted DRG neurons.

Our data suggest that βII spectrin functions, possibly in the context of the MPS, as a novel factor in TD-induced retrograde signaling and axon degeneration. Precisely how βII spectrin is involved in retrograde signaling in the axon remains to be determined. TD-induced retrograde signaling involves at least two pathways in sensory axons. The first is loss of pro-survival Akt activity, culminating in activation of the downstream transcription factor Foxo3a (*Simon et al., 2016*). The second is a gain of pro-degenerative MAPK signaling, mediated by DLK, and culminating in phosphorylation and activation of the transcription factor c-Jun (*Ghosh et al., 2011*; *Huntwork-Rodriguez et al., 2013*; *Simon et al., 2016*). In neurons lacking DLK, phosphorylation of c-Jun is blocked and

activation of Foxo3a is strongly attenuated (*Ghosh et al., 2011*; *Simon et al., 2016*), suggesting a crosstalk between these two pathways. Loss of βII spectrin does not block DLK activation following TD, but largely blocks phosphorylation of c-Jun and nuclear translocation of Foxo3a, suggesting that βII spectrin is potentially involved in both pathways, but the extent to which βII spectrin interacts with each remains to be determined.

## Structural plasticity of the MPS potentially functions to initiate retrograde signaling

Our data show that actin stabilization by Jasp prevents MPS disassembly and occludes the somatic response to TD (i.e. inhibits activation of DLK and c-Jun), suggesting that F-actin turnover is critical to this early phase of axon degeneration. Conversely, acutely destabilizing actin with LatA largely phenocopies TD, inducing MPS disassembly and promoting retrograde signaling (i.e. activation of DLK and c-Jun), culminating in activation of the somatic pro-degenerative program. βII spectrin depletion blocks the pro-degenerative effects of LatA-induced acute F-actin breakdown. Together with the observed early disassembly of MPS upon TD and the protection against TD-induced degeneration by βII spectrin depletion, these data suggest a possibility that structural plasticity of the MPS is important for pro-degenerative retrograde signaling.

The MPS lacks a net polarity toward the distal axon or cell body, so it is unlikely to act as a 'track' for pro-degenerative retrograde signal propagation. Instead, the MPS structure may possibly serve as a platform to coordinate the activation of this retrograde signal. The sub-membrane lattice structure of the MPS provides a mechanism for anchoring transmembrane proteins, likely in a way similar to the erythrocyte membrane skeleton (*Hoover and Bryant, 2000*; *Bennett and Baines, 2001*; *Baines et al., 2014*). Indeed, some functional membrane proteins are organized into periodic distributions along the axons by the MPS (*Xu et al., 2013*; *Zhong et al., 2014*; *D'Este et al., 2015*; *D'Este et al., 2017*). It is thus possible that the MPS sequesters and thus functionally inactivates some pro-degenerative signaling molecules in healthy axons to prevent aberrant signaling, but also primes axons for degeneration by recruiting these molecules to the appropriate region of the axons. MPS disassembly might then activate retrograde signaling by releasing these signaling molecules. On the other hand, inhibiting establishment of the MPS may lead to mislocalization of these signaling molecules, resulting in failure of retrograde signaling. This model could explain why stabilization of the MPS (with Jasplakinolide) or inhibition of its formation (with knockdown/knockout of βII spectrin) both blocks pro-degenerative retrograde signaling and axon degeneration, while MPS disassembly (by TD or LatA treatment), can promote retrograde signaling and axon degeneration. Another possibility is that MPS breakdown during TD (or following LatA treatment) produces a pool of soluble MPS components, such as spectrin and ankyrin, which could participate in retrograde trafficking. Of note, biochemical reconstitution of dynein-driven retrograde signaling in the axon has demonstrated a prominent role for PH domain-containing β spectrins, including βII spectrin, in bridging vesicular membranes to dynein-driven retrograde motors (*Muresan et al., 2001*). In addition, AnkB, which is associated with the MPS, has been shown as a motor protein adaptor important for retrograde transport of multiple organelles, such as endosomes and mitochondria, in hippocampal neurons (*Lorenzo et al., 2014*). The disassembly of the MPS to free these MPS components could promote the role of these proteins in retrograde transport, which might in turn facilitate retrograde signaling in TD-induced axon degeneration.

Finally, we note that it is difficult to disentangle the roles of different cytoskeleton structures due to their interactions and due to the fact that pharmacological and genetic manipulations often perturb multiple cytoskeletons simultaneously and may induce transcriptional responses (*Massaro et al., 2009*). Therefore, while acute manipulations that modify F-actin by LatA and Jasp have clear effects on the MPS, we cannot exclude the possibility that the observed effects of pharmacological manipulations on the initiation of axon degeneration occur through other actin structures. Similarly, the depletion of βII spectrin could also affect microtubule structure and microtubule-dependent retrograde transport in addition to the structure of the MPS (*Qu et al., 2017*). Accordingly, while our manipulations of F-actin and βII spectrin converge on the MPS, it is also possible that other cytoskeleton structures may contribute to the observed effects on axon degeneration.

# Materials and methods

**Key resources table**

| Reagent type (species) or resource | Designation | Source or reference | Identifiers | Additional information |
|---|---|---|---|---|
| Genetic reagent (*M. musculus*) | Sptbn1-flox | *Galiano et al., 2012* | RRID: MGI:5431637 | Dr. Matt Rasband (Baylor College of Medicine) |
| Genetic reagent (*M. musculus*) | Bax KO | *Knudson et al., 1995* | RRID: MGI:1857429 | Dr. Stanley Korsmeyer (via Jackson Labortory) |
| Antibody | mouse monoclonal anti-βII spectrin, clone 42 | Santa Cruz | Cat. # sc-136074; RRID: AB_2194501 | IF (1:100), WB (1:500) |
| Antibody | rabbit anti-Tubulin β 3 | BioLegend | Cat. # 845502; RRID: AB_2566589 | IF (1:2000) |
| Antibody | mouse monoclonal anti-Tubulin β 3 | BioLegend | Cat. # MMS-435P; RRID: AB_2313773 | IF (1:2000), WB (1:1000) |
| Antibody | rabbit anti-cleaved Caspase-3 | Cell Signaling | Cat. # 9661; RRID: AB_2341188 | IF (1:100) |
| Antibody | rabbit monoclonal anti-cleaved Caspase-3 | Cell Signaling | Cat. # 9664; RRID: AB_2070042 | IF (1:100) |
| Antibody | Rabbit monoclonal anti-phosphorylated c-Jun S73 | Cell Signaling | Cat. # 3270; RRID: AB_2129575 | IF (1:500) |
| Antibody | rabbit anti-Foxo3a | Cell Signaling | Cat. # 12829; RRID: AB_2636990 | IF (1:200) |
| Antibody | rabbit polyclonal anti-DLK | Genetex | Cat. # GTX124127 | WB (1:500) |
| Antibody | mouse anti-αII spectrin | BioLegend | Cat. # 803201; RRID: AB_2564660 | WB (1:250) |
| Antibody | rabbit anti-α-adducin | Abcam | Cat. # ab51130; RRID: AB_867519 | WB (1:100) |
| Antibody | mouse anti-Ankyrin B | Neuromab | Cat. # 75–145; RRID: AB_10673095 | WB (1:100) |
| Antibody | rabbit monoclonal anti-β actin | Revmab | Cat. # 31-1013-00; RRID: AB_2716368 | WB (1:3000) |
| Chemical compound, drug | Hoechst 33342 | ThermoFisher Scientific | Cat. # H3570 | IF (1:10,000) |
| Chemical compound, drug | Jasplakinolide | Millipore Sigma | Cat. # 420107 M | Drugs were pre-dissolved in culture medium (sonicate shortly if necessary) for MPS imaging and in DMSO for other assays. |
| Chemical compound, drug | Jasplakinolide | Abcam | Cat. # 141409 | |
| Chemical compound, drug | Latrunculin A | Millipore Sigma | Cat. # L5163 | |
| Chemical compound, drug | Latrunculin A | Abcam | Cat. # 144290 | |

*Continued on next page*

*Continued*

| Reagent type (species) or resource | Designation | Source or reference | Identifiers | Additional information |
|---|---|---|---|---|
| Chemical compound, drug | GNE-3511 | Millipore Sigma | Cat. # 533168 | |
| Chemical compound, drug | ABT-737 | Selleck Chem | Cat. # S1002 | |
| Commercial assay, kit | Caspase-Glo 3/7 reagent | Promega | Cat. # G8090 | |
| Transfected construct (adenovirus) | βII-spectrin silencing, Gift from Baylor College of Medicine (Matthew Rasband) | *Galiano et al., 2012* | | |
| Transfected construct (adenovirus) | Scrambled shRNA with GFP | Vector Biolabs | Cat. # 1122 | |
| Transfected construct (lentivirus) | Lentivirus expressing mouse Bcl-xL cDNA | *Simon et al., 2016* | | |
| Transfected construct (lentivirus) | Lentivirus expressing GFP DNA | *Simon et al., 2016* | | |
| Transfected construct (lentivirus) | Lentivirus expressing Cre recombinase | *Simon et al., 2016* | | |
| Transfected construct (lentivirus) | Lentivirus expressing cytoplasmic NMNAT1 cDNA | *Simon et al., 2016* | | |

## Mice

Animals were bred and used according to IACUC protocols at The Rockefeller University and Stanford University. Embryos were harvested from pregnant dams at stage E12.5 where the plug date is standardized as E0.5. Wild-type cultures were generated from CD1 mice (Charles River). For mutant strains, comparisons between wild-type and mutant used tissues derived from the embryos from a single pregnant female. *Bax* knockout mice (*Knudson et al., 1995*) were purchased from Jackson Labs. *Sptbn1* conditional knockout mice (*Zhang et al., 2013*) were a kind gift from Dr. Matthew Rasband (Baylor College of Medicine).

## Neuronal cultures

Sensory neuron cultures were established as previously described (*Simon et al., 2016*). Briefly, E12.5 DRGs were isolated from spinal cords and dissociated in 0.05% Trypsin-EDTA (Life Technologies) for 20 min, washed once in DMEM media containing 10% FBS, then resuspended in culture media: Neurobasal (Life Technologies) containing 2% (v/v) B27 (Life Technologies), 0.45% (v/v) Glucose, 2 mM glutamine, 100 U/ml penicillin, 100 μg/ml streptomycin, and supplemented with 50 ng/mL NGF (Promega). Cells were plated on dried substrate in multi-well chamber slides at a density of ~15,000 per 1–1.5 microliters, allowed to adhere in a humidified incubator for ~12–15 min, at which time the chambers were filled with media and cultures returned to the incubator. The plating configuration allows for radial axon outgrowth away from a pool of cell bodies located near the center of the well. For imaging proximal axons, we used dissociated cells uniformly plated at a reduced density of ~4000 per well to avoid axon bundling. Cells were treated with mitotic inhibitor (5 μM 5-fluorouracil and 5 μM uridine) the morning after plating. For experiments involving STORM imaging, E12.5 DRGs were dissected away from embryonic spinal cords and shipped overnight from either Rockefeller or Stanford to Harvard in Hibernate-E media (Life Technologies) supplemented with 2% (v/v) B27. Upon arrival the DRGs were dissociated and plated as above. All assays were performed at 7–8 days in vitro (DIV). In cases where viral manipulation was used, cultures were infected with virus at 1 DIV

and assayed at 8 DIV. Trophic deprivations were performed by replacing NGF-containing culture media with media containing no NGF as well as a function-blocking antibody to NGF. For STORM imaging experiments, cultures were plated on poly-D-lysine (40 µg/ml), and then N-cadherin (1 µg/ml, R and D systems) and fibronectin (5 µg/ml, R and D systems). All other cultures were plated on poly-D-lysine (100 µg/ml) and mouse natural laminin (10 µg/ml, Life Technologies).

## Viral production

Lentiviruses encoding GFP, Cre Recombinase, and cytoplasmic NMNAT1 were produced and used as previously described (*Simon et al., 2016*). AV-shRNA against βII spectrin was a kind gift from Dr. Matthew Rasband (Baylor College of Medicine) as described previously (*Hedstrom et al., 2008*; *Galiano et al., 2012*; *Zhong et al., 2014*). An AV- scrambled shRNA was used as a control (Vector Biolabs #1122).

## Western blotting

Protein abundance was determined by western blotting using standard techniques as previously described (*Simon et al., 2016*). For axon-specific lysates, cell bodies at the center of the well (spotted as described above) were removed with a scalpel and axons were lysed directly in the well in a buffer containing 50 mM Tris-HCl (pH 6.8), 8M Urea, 10% (w/v) SDS, 10 mM sodium EDTA, 50 mM DTT, supplemented with Brilliant Blue G. For each condition examined, axons harvested from at least eight spot cultures (15,000 neurons per spot) were pooled. Samples were resolved on 4–15% pre-cast protein gels (Bio-Rad #3450029) in all cases except for visualizing the molecular weight shift of DLK, for which a pre-cast 7.5% Super-Sep Phos-tag polyacrylamide gel (Wako, #198–17981) was used.

## Measurement of axonal DEVDase activity

Dissociated and re-aggregated sensory neuron spot cultures, prepared as described above, were grown in 24-well plastic cell culture plates and treated as indicated (TD, LatA, etc.). At the terminal time point in the assay, cell bodies were removed using a scalpel (identical to parallel cultures used for protein harvesting). Next, half the culture media was removed, followed by addition of an equal volume of Caspase-Glo 3/7 reagent (Promega #G810C). Following a one-hour lysis at room temperature with gentle rocking, the contents of each well was transferred to an opaque 96-well plate and luminescence was determined using a plate reader. A minimum of four independent spots were measured for each assay condition.

## cDNA production and Real-Time PCR measurements

Total RNA from eight spot cultures per condition was collected using the RNeasy Micro Kit (Qiagen # 74104). cDNA was generated using Superscript III (ThermoFisher #18-080-051) from a defined amount of total RNA (normalized across experimental conditions), followed by real-time PCR experiments using the Taqman Gene Expression system (ThermoFisher # 4369016). Pre-designed primer pairs were used for c-Jun (ThermoFisher #Mm00495062_s1) and GAPDH (ThermoFisher #Mm99999915_g1).

## Immunofluorescence and imaging

Cultures for visualization of c-Jun and Foxo3a were fixed in 4% paraformaldehyde and 10% sucrose in PBS for 15 min at room temperature. Following one wash in PBS the cultures were blocked in 6% donkey serum in PBS containing 0.1% Triton X-100 (PBS-X) for 3 hr, and incubated in primary antibody in 3% donkey serum in PBS-X, overnight at room temperature. The next morning, following three 10 min washes in PBS-X, AlexaFluor-conjugated secondary antibodies were added (1:500) in 3% donkey serum in PBS-X for 1 hr at room temperature. Following two 10-min washes in PBS-X, a final wash containing 1:1000 Hoechst dye in PBS-X was performed. Slides were mounted in Fluoromount-G (Southern Biotech) and imaged on a Nikon Eclipse upright microscope.

Cultures for STORM imaging and visualization of other MPS associated proteins were fixed at various DIV as indicated in the main text, at a final concentration of 4% (wt/vol) paraformaldehyde by adding equal volume of fixation buffer (8% wt/vol paraformaldehyde and 4% wt/vol sucrose in 1 × PBS) to cell culture medium. Neurons were fixed for 25 min, wash three times with 1 × PBS,

permeabilized with 0.05% (vol/vol) Triton X-100 in 1 × PBS for 5 min, blocked in blocking buffer (3% wt/vol BSA in 1 × PBS) for 50 min at room temperature and subsequently stained with either primary antibodies against βII spectrin and neuron-specific Class III β-tubulin (TuJ1) in blocking buffer for ~15 hr at 4°C. The samples were washed three times with 1 × PBS for 10 min each and then stained with secondary antibodies (Alexa Fluor 647 donkey anti-mouse, Invitrogen, A31571; Alexa Fluor 555 goat anti-rabbit, Invitrogen, A21428) in blocking buffer for 50 min – 1 hr at room temperature. The samples were then washed three times with 1 × PBS for 10 min each, fixed again with 4% (wt/vol) paraformaldehyde in 1 × PBS for 15 min, washed three times with 1 × PBS and stored in 1 × PBS before imaging.

The STORM setup was based on an Olympus IX-71 inverted optical microscope as described previously (*Huang et al., 2008*; *Rust et al., 2006*). A 405 nm (Coherent CUBE 405–50C) laser beam, a 560 nm (Coherent Sapphire) laser beam and a 640 nm (Coherent Genesis MX) laser beam were introduced into the sample through the back port of the microscope. A translation stage allowed the laser beams to be shifted toward the edge of the objective so that the emerging light reached the sample at incidence angles slightly smaller than the critical angle of the glass-water interface, thus illuminating only the fluorophores within a few micrometers of the coverslip surface. A T660LPXR (Chroma) dichroic mirror and an ET705/72M band-pass emission filter (Chroma) were used for imaging of Alexa Fluor 647; a Di01-561 (Semrock) dichroic mirror and an FF01-617/73 band-pass emission filter (Semrock) were used for imaging Alexa Fluor 555. For 3D STORM imaging, a cylindrical lens was inserted into the imaging path so that images of single molecules were elongated in $x$ and $y$ for molecules on the proximal and distal sides of the focal plane (relative to the objective), respectively (*Huang et al., 2008*).

The sample was imaged in an air-tight container filled with ~2.5 ml image buffer per well in a four-well chamber (10% (wt/vol) glucose, 50 mM Tris-HCl pH 7.5, additional 10 mM NaCl, 10 mM cysteamine, 1 mg/mL glucose oxidase (Sigma-Aldrich), and 40 µg/mL catalase (Roche Applied Science) in 1 × PBS).

Axons were selected based on TuJ1 staining (wide-field conventional image with 561 nm laser) to achieve a blind selection of regions for STORM imaging of βII spectrin and according to the criteria that they should be at between ~3 mm away from the cell body aggregation in the middle of the well to ~40 µm from the tip of an axon. Only unbundled axons with a diameter less than 700 nm were selected.

During STORM imaging, continuous illumination of 640 nm laser (~2 kW/cm2) was used to excite fluorescence from Alexa Flour 647 molecules and switched them into the dark state. Continuous illumination of the 405 nm laser was used to reactivate the fluorophores to the emitting state. The power of the activation lasers was adjusted during image acquisition so that, at any given instant, fluorophores in the emitting state were mostly non-overlapping with each other.

A typical STORM image was generated from a sequence of 25,000 image frames at a frame rate of 60 Hz. The recorded STORM movie was analyzed according to previously described methods (*Huang et al., 2008*; *Rust et al., 2006*). The centroid positions and ellipticities of the single-molecule images provided lateral and axial positions of each activated fluorescent molecule, respectively. Super-resolution images were reconstructed from the molecular coordinates by depicting each location as a 2D Gaussian peak.

## Quantification of STORM images

Autocorrelation analyses were performed on the molecule list of selected imaging areas similar to what has been described previously (*Zhong et al., 2014*; *He et al., 2016*; *Han et al., 2017*). Briefly, 1-D single-molecule localization distribution of an axon was first computed using a bin size of 10 nm. Then the distribution was segmented into several 1900 nm adjacent and non-overlapping regions to compute the 1-D spatial autocorrelation function of each region; the autocorrelation function of an axon was calculated by averaging that of all segmented regions in the axon. The autocorrelation amplitude of an axon was defined as the difference in autocorrelation values between the local maximum around 190 nm and the local minimum around 95 nm in the average autocorrelation function curve of the axon.

The Occupancy ratio was defined as the βII spectrin occupancy over TuJ1-marked regions. Briefly, conventional wide-field images of βII spectrin and TuJ1 were binarized and morphological operation was performed on binarized TuJ1 images to remove boundary regions without breaking objects

apart and convert axons into binary images of strokes. The strokes of axons were then registered with images of βII spectrin using an affine transform to correct for possible drift. βII spectrin signals outside of the TuJ1-marked regions were ignored. The occupancy ratio was calculated as the ratio of regions covered by βII spectrin signal over the TuJ1-marked regions.

## Phosphorylation analysis

In brief, 620 µg protein per replicate (n = 3), in 110 µl of 8M Urea, were trypsinized and desalted. 20 µg of each digest was subjected to protein profiling while the remainder was subjected to titanium dioxide based phosphopeptide enrichment as described previously (*Govek et al., 2018*). Samples were analyzed by high-resolution/high-accuracy LC-MS/MS (Q-Exactive Plus, ThermoFisher). LC setup was operated with a trap column. Tandem MS data were queried against Uniport's Mouse Complete Proteome (March 2016) using MaxQuant v. 1.6.0.13 (*Tyanova et al., 2016a*). Data were processed as a peptide centric stable isotope labeling by amino acids in cell culture experiment (Lysine +8 Da and Arginine +10 Da). Phosphorylation of Serine, Threonine and Tyrosine were allowed as variable modifications in addition to methionine oxidation and protein N-terminal acetylation. Only Peptide false discovery rate (1%) was used to filter matched peptides. Phosphorylation sites were considered if measured in 2-of-3 replicates and with a localization probability better than 0.75. Data were processed and analyzed using Perseus (*Tyanova et al., 2016b*). Phosphorylation sites were queried against PhosphoSite (*Hornbeck et al., 2015*). Measured phosphorylation sites of αII-spectrin, βII-spectrin, α-adducin, β−adducin and γ-adducin proteins are listed in *Figure 2—figure supplement 2* and quantified in *Figure 2—figure supplement 2—source data 1*. The mass spectrometry proteomics data have been deposited to the ProteomeXchange Consortium via the PRIDE (*Vizcaíno et al., 2016*) partner repository with the dataset identifier PXD009854.

## Statistical analysis

Statistical tests were performed using either MATLAB or Prism 7 (Prism). The statistical test applied to each quantification are indicated in each figure legend.

## Acknowledgements

The AV-based shRNA against βII spectrin and the βII spectrin conditional knockout were kind gifts of Dr. Matthew Rasband (Baylor College of Medicine).

# Additional information

### Funding

| Funder | Grant reference number | Author |
|---|---|---|
| National Institutes of Health | R35GM122487 | Xiaowei Zhuang |
| Howard Hughes Medical Institute | | Xiaowei Zhuang |
| National Institutes of Health | R01NS089786 | Marc Tessier-Lavigne |

The funders had no role in study design, data collection and interpretation, or the decision to submit the work for publication.

### Author contributions

Guiping Wang, Conceptualization, Data curation, Software, Formal analysis, Investigation, Visualization, Writing—original draft, Writing—review and editing; David J Simon, Conceptualization, Resources, Data curation, Formal analysis, Investigation, Visualization, Writing—original draft, Writing—review and editing; Zhuhao Wu, Conceptualization, Resources, Investigation, Writing—review and editing; Deanna M Belsky, Evan Heller, Melanie K O'Rourke, Guisheng Zhong, Investigation, Writing—review and editing; Nicholas T Hertz, Henrik Molina, Resources, Formal analysis, Investigation, Writing—review and editing; Marc Tessier-Lavigne, Xiaowei Zhuang, Conceptualization, Supervision,

Investigation, Funding acquisition, Writing—original draft, Project administration, Writing—review and editing

## Author ORCIDs
David J Simon ⓘD https://orcid.org/0000-0002-2683-5757
Zhuhao Wu ⓘD http://orcid.org/0000-0002-2471-0555
Xiaowei Zhuang ⓘD http://orcid.org/0000-0002-6034-7853

## Ethics
Animal experimentation: This study was performed in strict accordance with the recommendations in the Guide for the Care and Use of Laboratory Animals of the National Institutes of Health. Animals were bred and used according to IACUC protocols at The Rockefeller University and Stanford University. Work at Harvard was performed in accordance with the Guide for the Care and Use of Laboratory Animals of the National Institutes of Health. Mouse work at Stanford was performed under APLAC protocol 31688. Mouse work at Rockefeller was performed under IACUC protocol 14713. Mouse work at Harvard was performed under IACUC protocol 10-16-2.

## Decision letter and Author response
Decision letter https://doi.org/10.7554/eLife.38730.040
Author response https://doi.org/10.7554/eLife.38730.041

## Additional files

### Supplementary files
• Transparent reporting form
DOI: https://doi.org/10.7554/eLife.38730.036

### Data availability
The mass spectrometry proteomics data have been deposited to the ProteomeXchange Consortium via the PRIDE partner repository with the dataset identifier PXD009854.

The following dataset was generated:

| Author(s) | Year | Dataset title | Dataset URL | Database and Identifier |
|---|---|---|---|---|
| Nicholas T. Hertz, David J. Simon | 2019 | Structural plasticity of actin-spectrin membrane skeleton and functional role of actin and spectrin in axon degeneration | http://proteomecentral.proteomexchange.org/cgi/GetDataset?ID=PXD009854 | ProteomeXchange Consortium, PXD009854 |

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
