## [Decision Letter]

Thank you for submitting your article "Membrane cytoskeleton dynamics underlie the initiation of sensory axon degeneration" for consideration by *eLife*. Your article has been reviewed by three peer reviewers, and the evaluation has been overseen by a Reviewing Editor and Marianne Bronner as the Senior Editor. The following individual involved in review of your submission has agreed to reveal their identity: Vann Bennett (Reviewer #1)

The reviewers have discussed the reviews with one another and the Reviewing Editor has drafted this decision to help you prepare a revised submission.

Summary:

Wang and colleagues report several interesting observations connecting the recently discovered axonal spectrin/actin rings with signaling leading to axon degeneration in cultured sensory neurons following removal of NGF (a surrogate for physiological axon pruning during development). The periodicity of the rings is disrupted within a few hours of NGF withdrawal well before onset of apoptosis and even when apoptosis is prevented altogether. Multiple sites of phosphorylation of αII and βII-spectrin and adducin that occur following NGF deprivation are described, although this is not followed up. The effects of manipulating spectrin/actin rings are examined by targeting actin with pharmacological agents and by knocking down βII-spectrin.

Essential revisions:

The paper represents a set of intriguing and generally well-executed experiments, however it does not add up to a coherent story. The results are strong but somewhat conflicting. Stabilizing actin prevents degeneration, but so does eliminating βII-βspectrin. A possible (likely) resolution is that both actin and β-spectrin have roles in addition to assembly of spectrin-actin rings. One issue is that the lack of in vivo data. The study is a bit thin on mechanistic insight on (i) how NGF withdrawal leads to rapid MPS disassembly, and (ii) how MPS remodeling promotes retrograde degenerative events including DLK and c-Jun activation. The paper could be strengthened by addressing the following essential points:

1) Both actin and spectrin have pleiotropic roles that can complicate interpretation of inhibitors/knockdowns. A more precise target (such as a dedicated component of the spectrin/actin rings if one can be identified) could help. The authors should develop the unexpected finding that βII-spectrin elimination prevents/slows degeneration. Although inconvenient and difficult to explain in the short term, this could be the most interesting observation.

2) One suggested experiment is to induce degeneration with LatA in neurons that lack β-spectrin, according to the authors model the axons should be protected, as the actin depolymerization will not be able to transmit the signal to the cell body.

3) The authors provide evidence from a phosphoproteomic screen that MPS components may be phosphorylated following trophic deprivation and speculate that aberrant phosphorylation may lead to destabilization of MPS. Some direct evidence supporting this in sensory axons following TD would strengthen the study. Could the authors use phospho-deficient or phosphomimetic mutants of βII-spectrin to test their model? Furthermore, in a recent study (Unsain et al., 2018) that the authors cite, it was reported that NGF withdrawal results in a rapid 50% loss of βII-spectrin and actin in the MPS. Is this the case here? Does NGF withdrawal lead not only to aberrant phosphorylation but also a loss of MPS components? How can a rapid loss of MPS components following TD be reconciled with the proposed model here that MPS components are required to initiate axon degeneration?

4) In the absence of a clear-cut mechanism, the authors should provide stronger support for the link between rapid MPS disassembly and DLK phosphorylation/stability/retrograde transport by including more rigorous analyses similar to that reported by Larhammar et al., 2017. For example, quantification of DLK abundance in βII-spectrin null neurons, evidence of retrograde transport of DLK/P-*JNK* to cell bodies in mutant neurons, quantification of p-c-Jun results for Figures 3 and 5 rather than just showing images. Similarly, for the DLK1 western blots shown in Figures 3 and 4 supplements, there is no information if this is from a single experiment or representative of multiple analyses. Moreover, DLK phosphorylation should be probed by a phospho-specific DLK antibody if available. These analyses are necessary to provide more convincing support for the notion that βII-spectrin, an essential component of MPS, is required for well-characterized degenerative signaling events during trophic deprivation.

5) In the context of MPS disassembly initiating retrograde degenerative signaling, is the periodicity affected equally in all axonal segments after NGF withdrawal or does it show a specific pattern? Specifically, is there a progressive loss of periodicity from distal to proximal axon segments following NGF deprivation? It is not clear which regions of the axons were selected for imaging.

---

## [Author Response]

Essential revisions:The paper represents a set of intriguing and generally well-executed experiments, however it does not add up to a coherent story. The results are strong but somewhat conflicting. Stabilizing actin prevents degeneration, but so does eliminating βII-spectrin. A possible (likely) resolution is that both actin and β-spectrin have roles in addition to assembly of spectrin-actin rings. One issue is that the lack of in vivo data. The study is a bit thin on mechanistic insight on (i) how NGF withdrawal leads to rapid MPS disassembly, and (ii) how MPS remodeling promotes retrograde degenerative events including DLK and c-Jun activation.

We thank the editor and reviewers for their constructive comments. Based on these comments, we have performed additional experiments and analyses and made text revisions, which have strengthened our manuscript.

Most of the comments in this general statement have been elaborated in the specific points below and hence we will address these comments in our response to the specific comments below. Here we only address the comment “One issue is that the lack of in vivo data” which has not been further mentioned below.

We agree that studying axon degeneration in vivo is an important goal. However, at any given time in vivo, of all axons, which travel in tight bundles of several hundred axons, only a small subset is degenerating, which has prevented us and other labs from having the spatial resolution to observe these few dying axons in the background of many more healthy ones. Indeed, we developed the whole-mount immunolabelling protocol iDISCO in part to attempt to visualize sensory axon degeneration in vivo, but have thus far been unable to visualize this process.

Hence, the use of cultured primary neurons to study the detailed molecular mechanisms of axon degeneration is the general practice in the field, particularly because, by initiating NGF deprivation at a fixed time we are able to see near-synchronous degeneration of axons. It is worth noting that we and others have extensively documented that the pathways of axon degeneration uncovered in vitro do indeed relate to the ultimate survival/death of these axons in vivo. For example, mouse mutants that delay (*Puma, DR6, APP, Puma*) or block (*Bax, Caspase-3*) axon degeneration display an increase in un-pruned axons in the visual system and an excess of sensory axons projecting to the periphery at later embryonic stages (Nikolaev et al., 2009; Simon et al., 2012, Olsen et al., J.Neurosci., 2014; Maor-Nof et al., 2016).

Finally, an additional challenge for this study is that super-resolution imaging in vivo has also been generally challenging. Moreover, the tight bundling of sensory axons in vivo (see above) would further hinder our ability to efficiently visualize the MPS and its disruption in the subset of axons that degenerate at any moment using super-resolution imaging.

The paper could be strengthened by addressing the following essential points:1) Both actin and spectrin have pleiotropic roles that can complicate interpretation of inhibitors/knockdowns. A more precise target (such as a dedicated component of the spectrin/actin rings if one can be identified) could help. The authors should develop the unexpected finding that βII-spectrin elimination prevents/slows degeneration. Although inconvenient and difficult to explain in the short term, this could be the most interesting observation.

We agree with the reviewers that both actin and spectrin have pleiotropic roles that can complicate the interpretation of pharmacological actin manipulations or knockdown/knockout of βII spectrin. It would indeed be great to find more precise targets that only affect the MPS. To this end, we have made substantial effort trying to achieve this goal:

a) We tried generating a mutant mouse where the actin-binding domain of βII spectrin can be conditionally deleted in culture using Cre-Lox recombination, with the hope of disrupting the MPS in more specific ways while still maintaining the presence of βII spectrin proteins and actin filaments. We tried this conditional knockout strategy because we are concerned that germline knockout would be lethal to the animal. However, even when we infected the culture neurons from this mutant mouse with a very high dose of Cre virus, only a fraction of βII spectrin had the actin-binding domain successfully deleted, as assayed by western blot. The problem of insufficient deletion could be either a result of low recombination rate for distantly spaced LoxP sites (the actin-binding domain encompasses 73 kb of genomic DNA) or due to the un-degraded full-length βII spectrin produced before the incidence of recombination. While we are exploring means to increase the recombination efficiency, these approaches require significantly more time than the scope of this revision. We plan to follow up on this in our future work.

b) We also generated a germline knock-in mouse model with the K2207Q mutation in the PH domain of βII-spectrin which has been shown to abolish binding affinity of βII-spectrin for phosphoinositides in MDCK cells (He et al., J Cell Biol 2014 Jul 21; 206(2)). This mutant also has the benefit of maintaining the original length of βII-spectrin. Our logic was that, by abolishing the binding to phosphoinositides, we could potentially remove βII-spectrin from the plasma membrane and thus disrupt the MPS. However, we found that the MPS still persisted along axons, albeit with a discontinuous distribution containing gaps devoid of the MPS. We also did not observe protection against TD-induced degeneration in these cultures, not unexpectedly because the MPS is still present in axons.

c) We observed TD-associated phosphorylation of MPS components, which could potentially facilitate dissociation of the MPS. Hence, it is possible that the MPS structure may be disrupted by engineering phospho-mimetic (S/T>D) amino acids into the sequence of βII spectrin. We therefore generated a panel of S/T>D (phospho-mimetic) and S/T>A (phospho-deficient) mutations into a cDNA encoding the βII spectrin (*Sptbn1*) gene. However, *Sptbn1* cDNA is ~7.5kb long, far exceeding the maximum packaging sizes for both lentivirus and AAV, which has prevented us from successfully introducing these constructs to DRG neurons. We also made several attempts to introduce these mutations by cDNA transfection, each of which resulted in toxicity. Similar toxicity has been described for cDNA transfection in DRGs by other groups (Yu et al., Front Mol Neurosci. 2015 Feb 2;8:2.). In addition, we have explored adenovirus as a viral delivery system, which can typically package open reading frames up to ~6kb, and unfortunately these efforts also failed to express the full-length product of this cDNA in neurons.

Therefore, despite several months of effort, we were not able to generate mutants that can completely disentangle the role of the MPS from the role of other cytoskeletal structures or isolated βII spectrin. Therefore, even though the simplest explanation of our observed effects of actin and spectrin manipulations converge on the MPS, we revised the Discussion section of the manuscript to explicitly acknowledge that other cytoskeleton structures or other roles of actin and spectrin outside the MPS may also contribute to the observed effects of actin manipulations and spectrin perturbation on axon degeneration (see subsection “Structural plasticity of the MPS potentially functions to initiate retrograde signaling”, last paragraph).

We also agree with the reviewers that it would be interesting to develop the unexpected finding that βII spectrin elimination prevents/slows degeneration. Following the reviewer’s suggestion, we have performed additional experiments on retrograde signaling induced by TD in βII spectrin knockdown neurons to show that the loss-of-βII spectrin acts downstream of the activation of DLK kinase but upstream of the activation of the transcription factor c-Jun (see subsection “Depletion of MPS component βII spectrin protects axons from degeneration”, first paragraph and Figure 5, Figure 5—figure supplement 1). We have also performed additional experiments on LatA-induced axon degeneration in βII spectrin knockdown neurons to show that βII spectrin depletion also protect axons from degeneration under LatA treatment by measuring the Caspase3-like enzymatic activity (‘DEVDase’) in axons (see subsection “Depletion of the MPS component βII spectrin protects axons from degeneration”, last paragraph; Figure 5—figure supplement 3). Finally, we elaborated the implication of the finding that βII spectrin depletion protect axons from degeneration by providing two possible models to explain this result: 1) It is possible that the MPS sequesters pro-degenerative signaling molecules in healthy axons to prevent aberrant signaling, but also primes axons for degeneration by recruiting these molecules to the appropriate region of the axons. MPS disassembly might then activate the retrograde signaling by releasing these signaling molecules. On the other hand, inhibition of the formation of the MPS by βII spectrin depletion could lead to mis-localization of these signaling molecules, resulting in failure of retrograde signaling. 2) Another possibility is that MPS breakdown during TD (or following LatA treatment) produces a pool of soluble MPS components, such as spectrin and ankyrin, which could participate in retrograde trafficking of pro-degenerative signals. Both spectrin and ankyrin B have been shown to interact with motor proteins that move on microtubules. The disassembly of the MPS to free these MPS components could promote the role of these proteins in retrograde transport, which may in turn facilitate retrograde signaling in TD induced axon degeneration. Please see subsection “Structural plasticity of the MPS potentially functions to initiate retrograde signaling”, second paragraph for these discussions.

2) One suggested experiment is to induce degeneration with LatA in neurons that lack β-spectrin, according to the authors model the axons should be protected, as the actin depolymerization will not be able to transmit the signal to the cell body.

We thank the reviewers for suggesting this informative experiment. We have performed this additional experiment and added the results to the revised manuscript. As suggested by the reviewer, our model predicts that βII spectrin deletion will protect axon from degeneration induced by LatA. This is indeed what we observed. We elaborate this finding below:

In our original manuscript we showed that application of LatA led to activation of axonal Caspase-3 and that axonal Caspase-3 activation does not occur when LatA is added to isolated axons that have been severed from their cell bodies (Figure 3E). These results indicate that acute breakdown of F-actin in the axon generates a retrograde signal to the cell body, followed by an anterograde signal from the cell body to the axon that activates axonal Caspase-3, similar to the mechanism by which TD activates axonal Caspase-3.

As suggested by the reviewers, we now performed new experiments to study whether LatA treatment still induces axon degeneration in βII spectrin knockdown neurons. Our new results showed that βII spectrin knockdown indeed prevented increase in axonal Caspase-3-like enzymatic activity in this case, similar to the case of TD-induced axon degeneration (see subsection “Depletion of the MPS component βII spectrin protects axons from degeneration”, last paragraph, Figure 5—figure supplement 3A). We note that βII spectrin knockdown did not prevent c-Jun activation in the case of LatA treatment (see the aforementioned paragraph, Figure 5—figure supplement 3B, C), even though TD-induced c-Jun activation was prevented by βII spectrin knockdown. This is likely because LatA treatment, which disrupts all forms of F-actin, initiates additional, less specific and βII spectrin-independent signal to activate c-Jun, which does not occur during TD. Nevertheless, the observation that βII spectrin knockdown blocks both LatA-induced and TD-induced axonal Caspase-3-like enzymatic activity indicates that βII spectrin is critical to retrograde signaling on the TD-induced (and related LatA-induced) axon degeneration pathway.

3) The authors provide evidence from a phosphoproteomic screen that MPS components may be phosphorylated following trophic deprivation and speculate that aberrant phosphorylation may lead to destabilization of MPS. Some direct evidence supporting this in sensory axons following TD would strengthen the study. Could the authors use phospho-deficient or phosphomimetic mutants of βII-spectrin to test their model?

We agree that it would be very interesting to obtain direct evidence supporting the hypothesis that TD-induced phosphorylation of MPS component may lead to destabilization of MPS. We have made several attempts to address this issue by generating both phospho-deficient (S/T>A) and phospho-mimetic (S/T > D) mutation variants of βII spectrin. However, the βII spectrin (*Sptbn1*) gene, which is ~7.5kb long, far exceeds the maximum packaging sizes for both lentivirus and AAV, which has prevented us from successfully introducing these constructs to DRG neurons. We made several attempts on introducing these mutations by cDNA transfection, each of which resulted in toxicity. Similar toxicity has been described for cDNA transfection in DRGs by other groups (Yu et al., Front Mol Neurosci. 2015 Feb 2;8:2.). In addition, we have explored adenovirus as a viral delivery system, which can typically package open reading frames up to ~6kb. Unfortunately, these efforts also failed to express the full-length product of this cDNA in neurons. We therefore were not able to perform this suggested experiment despite several months of effort. Because we have not been able to address the functional role for these phosphorylations, we have further softened our discussion regarding their relevance to MPS disruption and axon degeneration (see subsection “The early disassembly of MPS is independent of apoptotic pathway activation and protein loss”, last paragraph and subsection “Depletion of βII-spectrin inhibits retrograde signaling and protects axons from degeneration”). Despite this, we feel that the inclusion of the identification of these sites has value and will be of great interest to labs that are exploring the biology of the MPS.

Furthermore, in a recent study (Unsain et al., 2018) that the authors cite, it was reported that NGF withdrawal results in a rapid 50% loss of βII-spectrin and actin in the MPS. Is this the case here? Does NGF withdrawal lead not only to aberrant phosphorylation but also a loss of MPS components? How can a rapid loss of MPS components following TD be reconciled with the proposed model here that MPS components are required to initiate axon degeneration?

We note that (Unsain et al., 2018) did not probe the abundance of βII-spectrin, actin or other MPS components. Instead, Unsain et al. reported the so-called MPS abundance, which is defined as the fraction of axonal regions that exhibited periodic distribution of βII-spectrin or actin. Hence, this study only concluded that the MPS structure is disrupted upon NGF withdraw, but did not conclude on any loss of βII-spectrin, actin or other MPS components.

In our manuscript, we showed that the MPS disassembly upon TD occurs early and prior to the activation of the apoptotic activity and that the TD-induced MPS disassembly still occurs in axons where the apoptotic pathway is blocked. These results suggest that the observed MPS disassembly is unlikely a result of the reduced abundance of MPS components.

However, in light of this reviewer comment, we performed additional experiments to probe whether there is a decrease in abundance of the MPS molecular components upon TD.

a) We measured the total abundance of βII spectrin in axons, reflected by the average fluorescence intensity, and found that the βII spectrin remains unchanged during the entire 12 h observation period after TD (Figure 2—figure supplement 1), whereas MPS disassembly has already occurred substantially before the 12 h time point (MPS is substantially disrupted already by 3-6 h time after TD, see Figure 1, 2).

b) We extended our analysis to test whether other MPS components are lost during the relevant window of TD. Towards this end, we performed a series of Western Blots on axonal proteins to examine the abundance of MPS components, αII spectrin, βII spectrin, and actin, during a time course of TD.

Our key observations (Figure 2B) are:

i) Total protein abundance of βII spectrin and actin remained constant during the entire 18hr assay window.

ii) Proteolysis of αII spectrin did occur during TD, but only beginning at the 12hr time point and in a Caspase-dependent manner.

In sum, the behavior of the MPS that we observed is not a consequence of reduced abundance of MPS component proteins. These data additionally contribute towards our hypothesis that MPS disassembly potentially has an upstream role during axon degeneration.

4) In the absence of a clear-cut mechanism, the authors should provide stronger support for the link between rapid MPS disassembly and DLK phosphorylation/stability/retrograde transport by including more rigorous analyses similar to that reported by Larhammar et al., 2017. For example, quantification of DLK abundance in βII-spectrin null neurons, evidence of retrograde transport of DLK/P-JNK to cell bodies in mutant neurons, quantification of p-c-Jun results for Figures 3 and 5 rather than just showing images. Similarly, for the DLK1 western blots shown in Figures 3 and 4 supplements, there is no information if this is from a single experiment or representative of multiple analyses. Moreover, DLK phosphorylation should be probed by a phospho-specific DLK antibody if available. These analyses are necessary to provide more convincing support for the notion that βII-spectrin, an essential component of MPS, is required for well-characterized degenerative signaling events during trophic deprivation.

As suggested, we performed additional experiments to examine the DLK abundance in βII spectrin null axons and found that it was not appreciably different than that in WT axons (See Figure 5I). Interestingly, although loss of βII spectrin protected axons from degeneration and inhibited the activation of c-Jun, it did not block DLK activation (assayed by the phosphorylation-dependent shift in DLK molecular weight) (Figure 5I), suggesting that βII spectrin likely functions downstream of DLK activation but upstream of c-Jun.

Also as suggested by the reviewer, we have added quantification of the magnitude of c-Jun phosphorylation in Figures 3 and 5 to show more clearly the c-Jun activation in TD cultures and LatA treated cultures and the reduction in c-Jun activation with DLK inhibition and βII spectrin depletion. In addition to quantifying phosphorylated c-Jun, we also added an additional assay to quantify increase in c-Jun mRNA abundance induced by TD. This induction is also largely blocked by βII spectrin knockdown (Figure 5—figure supplement 1D).

We thank the reviewers for the suggestion to include information about our sample size. We have updated our figure legends of Figure 3 and Figure 4 supplements (Figure 3—figure supplement 1 and Figure 4—figure supplement 2) to include the number of DLK1 western blot experiments performed for each representative figure shown (3 independent experiments).

Regarding the reviewer’s comment about detection of DLK activation: DLK is activated by phosphorylation near its active site and unfortunately no phospho-specific antibodies to these sites have been published or are commercially available. There are phosphospecific antibodies to phosphorylation sites outside of the activation site that work in 293T cells (Huntwork-Rodriguez et al., 2013), but we have been unable to obtain a convincing signal in our axonal samples using these antibodies. Nevertheless, the molecular weight shift that we employed is both widely used and well-validated as a DLK activation assay in the previous studies (Watkins et al., 2013; Huntwork-Rodriguez et al., 2013; Larhammar et al., 2017; Larhammar et al., J.Neurosci., 2017), and therefore we are confident in its use as a surrogate for DLK activation.

5) In the context of MPS disassembly initiating retrograde degenerative signaling, is the periodicity affected equally in all axonal segments after NGF withdrawal or does it show a specific pattern? Specifically, is there a progressive loss of periodicity from distal to proximal axon segments following NGF deprivation? It is not clear which regions of the axons were selected for imaging.

We apologize for not making this clear in our manuscript. Our results here focus on the distal axons. In light of this reviewer comment, we have performed additional experiments to image the proximal axon segments adjacent to the cell body, but interestingly, the MPS is not as well formed (i.e. βII spectrin exhibits relatively poor periodicity) in the axon region proximal to the cell body of DRG neurons even in the presence of NGF (Figure 1—figure supplement 2). We have now included this result in the revised manuscript. In the distal axons, without the presence of the cell body in the same field of view, it is difficult to determine which end of the axon segment is closer to the cell body. In general, the density of the axons makes it challenging to trace the axons all the way back to the cell body to determine orientation of the axon segments, which prevents us from making a conclusion on whether the disruption of the MPS propagates from distal to proximal axon segments.